



# Long-term observations of atmospheric aerosol, cloud condensation nuclei concentration and hygroscopicity in the Amazon rain forest – Part 1: Size-resolved characterization and new model parameterizations for CCN prediction

Mira L. Pöhlker[1], Christopher Pöhlker[1], Thomas Klimach[1], Isabella Hrabe de Angelis[1], Henrique M. J. Barbosa[2], Joel Brito[2,a], Samara Carbone[2], Yafang Cheng[1], Xuguang Chi[1,3], Florian Ditas[1], Reiner Ditz[1], Sachin S. Gunthe[4], Jürgen Kesselmeier[1], Tobias Könemann[1], Jošt V. Lavrič[5], Scot T. Martin[6], Daniel Moran-Zuloaga[1], Diana Rose[7], Jorge Saturno[1], Hang Su[1], Ryan Thalman[8,b], David Walter[1], Jian Wang[8], Stefan Wolff[1,9], Paulo Artaxo[2], Meinrat O. Andreae[1], and Ulrich Pöschl[1]

[1]Multiphase Chemistry and Biogeochemistry Departments, Max Planck Institute for Chemistry, 55020 Mainz, Germany.

[2] Institute of Physics, University of São Paulo, São Paulo 05508-900, Brazil.

[3] Institute for Climate and Global Change Research & School of Atmospheric Sciences, Nanjing University, Nanjing, 210093, China.

[4] EWRE Division, Department of Civil Engineering, Indian Institute of Technology Madras, Chennai 600036, India.

[5] Department of Biogeochemical Systems, Max Planck Institute for Biogeochemistry, 07701 Jena, Germany.

[6] School of Engineering and Applied Sciences, Harvard University, Cambridge, MA 02138, USA.

[7] Institute for Atmospheric and Environmental Research, Goethe University Frankfurt am Main, 60438 Frankfurt, Germany.

[8] Biological, Environmental & Climate Sciences Department, Brookhaven National Laboratory, Upton, NY 11973-5000, USA.

[9] Instituto Nacional de Pesquisas da Amazonia (INPA), Manaus-AM, CEP 69083-000, Brazil.

[a] now at: Laboratory for Meteorological Physics, University Blaise Pascal, Clermont-Ferrand, France.

[b] now at: Department of Chemistry, Snow College, Richfield, UT 84701, USA

*Correspondence to:* Mira L. Pöhlker (m.pohlker@mpic.de)





**Abstract.** Size-resolved long-term measurements of atmospheric aerosol and cloud condensation nuclei (CCN) concentrations as well as hygroscopicity were conducted at the remote Amazon Tall Tower Observatory (ATTO) in the central Amazon Basin over a one-year period and full seasonal cycle (March 2014 - February 2015). The presented measurements provide a climatology of CCN properties for a characteristic central Amazonian rain
forest site.

The CCN measurements were continuously cycled through 10 levels of supersaturation ($S = 0.11$ to $1.10\,\%$) and span the aerosol particle size range from 20 to 245 nm. The observed mean critical diameters of CCN activation range from 43 nm at $S = 1.10\,\%$ to 172 nm at $S = 0.11\,\%$. The particle hygroscopicity exhibits a pronounced size dependence with lower values for the Aitken mode ($\kappa_{\mathrm{Ait}} = 0.14 \pm 0.03$), elevated values for the
accumulation mode ($\kappa_{\mathrm{Acc}} = 0.22 \pm 0.05$), and an overall mean value of $\kappa_{\mathrm{mean}} = 0.17 \pm 0.06$, consistent with high fractions of organic aerosol.

The hygroscopicity parameter $\kappa$ exhibits remarkably little temporal variability: no pronounced diurnal cycles, weak seasonal trends, and few short-term variations during long-range transport events. In contrast, the CCN number concentrations exhibit a pronounced seasonal cycle, tracking the pollution-related seasonality in total
aerosol concentration. We find that the variability in the CCN concentrations in the central Amazon is mostly driven by aerosol particle number concentration and size distribution, while variations in aerosol hygroscopicity and chemical composition matter only during a few episodes.

For modelling purposes, we compare different approaches of predicting CCN number concentration and present a novel parameterization, which allows accurate CCN predictions based on a small set of input data.



## 1 Introduction

### 1.1 Atmospheric aerosols and clouds

In our current understanding of the Earth's climate system and its man-made perturbation, the multiscale and feedback-rich life cycles of clouds represent one of the largest uncertainties (Stevens et al., 2016; Boucher et al.,

2013). Accordingly, the adequate and robust representation of cloud properties is an Achilles' heel in climate modelling efforts (Bony et al., 2015). Atmospheric aerosols are a key ingredient in the life cycle of clouds (known as aerosol indirect effect) as they affect their formation, development, and properties by acting as cloud condensation nuclei (CCN) and ice nuclei (IN) (Lohmann and Feichter, 2005; Rosenfeld et al., 2008). Aerosol particles can originate from various natural and anthropogenic sources and span wide ranges of concentration,

particle size, composition, as well as chemical and physical properties (Pöschl, 2005). Their activation into cloud droplets depends on their size, composition, and mixing state as well as the water vapor supersaturation (e.g., Dusek et al., 2006; McFiggans et al., 2006; Andreae and Rosenfeld, 2008; Su et al., 2010; Köhler, 1936). The microphysical link between clouds and aerosol has been subject of manifold and long-term research efforts. On one hand, the cycling of CCN as well as their relationship to the aerosol population has been studied in a variety

of field experiments worldwide (e.g., Jurányi et al., 2011; Paramonov et al., 2015; Rose et al., 2010; Gunthe et al., 2009). On the other hand, the knowledge obtained from the growing body of field data has been translated into different parametrization strategies that represent the cloud-aerosol microphysical processes in modelling studies (e.g., Petters and Kreidenweis, 2007; Su et al., 2010; Mikhailov et al., 2013; Nenes and Seinfeld, 2003; Deng et al., 2013).

### 1.2 Amazon rain forest and its hydrological cycle

The Amazon rain forest is a unique and important ecosystem for various reasons, such as its high density and diversity of life, its role as major carbon storage, and its large recycling rate of energy and water in the Earth's hydrological cycle (Gloor et al., 2015; Brienen et al., 2015; Olivares et al., 2015; Yanez-Serrano et al., 2015). In

times of global change, the man-made disturbance and pressure on this ecosystem have strongly increased and have started a transition of the Amazon into an uncharted future (Davidson et al., 2012; Lawrence and Vandecar, 2015). In the context of atmospheric composition, the Amazon is unique since it represents one of the last terrestrial locations worldwide that allows – at least for part of the year – to investigate an relatively undisturbed state of the atmosphere in the absence of major anthropogenic pollution (Andreae, 2007; Hamilton et al., 2014; Andreae et al.,

2012; Roberts et al., 2001).

Overall, the troposphere over the Amazon is defined by the alternation of a relatively clean wet season and a polluted dry season, as outlined in more detail in previous studies (e.g., Martin et al., 2010b; Andreae et al., 2015; Mishra et al., 2015; Andreae et al., 2012). In this manuscript, we use the following classification of the Amazonian seasons[1]: (i) the *wet* season typically spans February to May and shows the cleanest atmospheric state, (ii) the

---

[1] Note that this definition of the seasons in the central Amazon is oriented on the seasonality in aerosol sources and prevalence rather than the meteorological conditions. For example, the 'meteorological wet season' typically has its core period in February (maximum in precipitation), whereas the 'pollution-defined wet season' has its core period in April/May (e.g., minimum in CO and BC concentrations) (Andreae et al, 2015).



*transition period from wet to dry* season typically spans June and July, (iii) the *dry* season months August to November show the highest pollution levels, and (iv) the *transition period from dry to wet* season spans December and January (Andreae et al., 2015; Moran-Zuloaga et al., 2016).

A lively discussed aspect of the Amazonian hydrological cycle is the potential impact of changing aerosol regimes, which oscillate between polluted and pristine extremes, on the development of clouds and precipitation (e.g., Rosenfeld et al., 2008; Andreae et al., 2004; Roberts et al., 2003). A variety of pollution-induced changes in cloud properties, such as increased cloud drop concentrations with a corresponding decrease of their average size, intense competition for water vapor and thus a deceleration of drop growth rates, suppression of supersaturation, reduced coalescence of smaller droplets, increased cloud depths as well as an invigoration of cloud dynamics and rain, are well documented (e.g., Freud et al., 2008; Koren et al., 2004; Koren et al., 2012).

Overall, the aforementioned observations indicate that increasing aerosol concentrations can have substantial impacts on spatial and temporal rainfall patterns in the Amazon (e.g., Martins et al., 2009a; Reutter et al., 2009). In view of the globally increasing pollution levels and the ongoing deforestation in the Amazon, pollution-triggered perturbations of the hydrological cycle are discussed as potential major threats to the Amazonian ecosystem, its forest structure, stability, and integrity (e.g., Coe et al., 2013; Junk, 2013).

### 1.3 Previous CCN measurements in the Amazon

Ground-based and airborne CCN measurements have been conducted in a number of field campaigns in the Amazon Basin as outlined below in chronological order, constituting the baseline and context for the present study.

1998: Roberts et al. (Roberts et al., 2001; Roberts et al., 2002) have conducted the first CCN measurements in the Amazon in the context of the LBA/CLAIRE-98 campaign (ground-based, Balbina site, March and April 1998) and pointed out that under clean conditions the CCN concentration $N_{CCN}(S)$ (at a certain supersaturation $S$) in the "Green Ocean" Amazon is surprisingly similar to conditions in the maritime "Blue Ocean" atmosphere. Regarding the low natural $N_{CCN}(S)$, which is dominated by mostly organic particles, they further suggested that cloud and precipitation properties may react sensitively to pollution-induced increases of the total aerosol load.

1999: In the context of the LBA-EUSTACH campaign in 1999, ground-based CCN measurements at three different sites in the Amazon Basin have been conducted (Andreae et al., 2002; Roberts et al., 2003). This was the first study on CCN properties and cloud dynamics under the influence of strong biomass burning emissions in the Amazon.

2001: In the follow-up study LBA/CLAIRE-2001 in July 2001, ground-based (Balbina site) and airborne measurements (around Manaus) have been conducted. For the ground-based study, Rissler et al. (2004) combined hygroscopicity tandem differential mobility analyzer (HTDMA) with CCN measurements, focusing on the CCN-relevant water soluble fraction in the particles, and provided a CCN closure and parametrization for model approaches. In addition, an airborne analysis of the aerosol and CCN properties has been conducted, focusing on the contrast between the Amazonian background air and the Manaus plume (Kuhn et al., 2010).

2002: Subsequently, in the course of the LBA-SMOCC campaign in Southern Brazil during major biomass burning episodes (Rondônia state, September and October 2002), ground-based and airborne CCN



measurements have been performed (Martins et al., 2009b; Vestin et al., 2007). A major finding of this study has been that the CCN efficiency of natural biogenic and manmade pyrogenic (cloud-processed) aerosols is surprisingly similar (Andreae et al., 2004). Furthermore, $N_{CCN}(0.5\ \%)$ was found as a valuable predictor for the required cloud depth of warm rain formation, which is an important property for the cloud dynamics (Freud et al., 2008).

2008: In the context of the AMAZE-08 campaign (ground-based, ZF2 site, February and March 2008), the first size-resolved CCN measurements in the Amazon have been conducted (Gunthe et al., 2009; Martin et al., 2010a). These studies report that aerosol particles in the Aitken and accumulation modes, which represent the CCN-relevant size range, predominantly contain organic constituents and thus reveal comparably low hygroscopicity levels. The observed hygroscopicity parameter $\kappa$ ranges between 0.1-0.2, which corresponds with the typical hygroscopicity of secondary organic aerosol (SOA) (Andreae and Rosenfeld, 2008)).

2010/11: During several short observational periods, Almeida et al. (2014) measured total CCN concentrations around the city of Fortaleza in northeast Brazil. The selected measurement locations receive wind from changing directions. Accordingly, the response of the CCN population to marine, urban, and rural air masses has been investigated.

2013: Recently, Whitehead et al. (2016) have reported results from further short-term, size-resolved CCN and HTDMA measurements that were conducted north of Manaus (ground-based, ZF2 site, July 2013) as part of the Brazil-UK Network for investigation of Amazonian atmospheric composition and impacts on climate (BUNIAACIC) project. The results of this study agree well with Gunthe et al. (2009).

2014/15: In the context of the international field campaign observation and modeling of the Green Ocean Amazon (GoAmazon2014/5), size-resolved CCN measurements have been conducted at three sites in and around Manaus: the ATTO site (T0a, pristine rain forest), which is discussed in the present study, the T2 site (in Manaus, urban environment), and the T3 site (rural site in the Manaus plume) (Martin et al., 2016; Thalman et al., 2016). All three size-resolved CCN measurements in the context of GoAmazon2014/5 took place in close collaboration. Moreover, CCN measurements were conducted onboard of the G-1 aircraft during the GoAmazon2014/5 intensive observation periods IOP1 and IOP2 (Martin et al., 2016).

2014: Furthermore, as part of the German-Brazilian ACRIDICON (Wendisch et al., 2016) and CHUVA (Machado et al., 2014) projects, airborne CCN measurements have been conducted over the entire Amazon Basin (September 2014). The results of this study are currently being analyzed for an upcoming publication and represent an ideal complement to the long-term data of the present study.

In addition to the aforementioned CCN measurements, some further studies relied on HTDMA measurements to probe the aerosol hygroscopicity and particle growth factors below 100 % RH, which can be used to extrapolate the CCN activity in supersaturation regimes (Rissler et al., 2006; Zhou et al., 2002).



### 1.4 Aims and scope of this study

All of the previously published CCN measurements in the Amazon have been conducted over relatively short time periods up to several weeks. In addition, size-resolved CCN measurements are still sparse in the Amazon region. In this study, we present the first continuous, long-term, and size-resolved CCN data set from the Amazon Basin, which spans a full seasonal cycle and therefore represents the CCN properties during contrasting seasonal conditions.

The focus of this study is on presenting major trends and characteristics of the CCN population in the Amazon Basin. Thus, our study contributes to a global inventory of CCN properties, representing this unique and climatically important ecosystem. We extract key CCN properties and parameters that help to include CCN predictions in the Amazon region into future modeling studies. Based on the dataset, different parametrization strategies for CCN prediction are compared and discussed for the present data. Moreover, a novel and generalized CCN parametrization is presented, which allows efficient modelling of CCN concentrations based on a minimal set of basic aerosol properties.

This manuscript represents part 1 of a comprehensive analysis of the CCN cycling in the central Amazon. It covers the overall trends and presents annually averaged CCN parameters as well as characteristic differences in the CCN population between the Amazonian seasons. A companion paper (part 2) provides in-depth analyses of particularly interesting events through short-term case studies and aims for a more emission- and process-related understanding of the CCN variability (M. L. Pöhlker et al., 2016b).

### 2 Methods

### 2.1 Measurement site and period

The measurements reported in this study were conducted at the Amazon Tall Tower Observatory (ATTO) site (S 02° 08.602', W 59° 00.033', 130 m a.s.l.), which is located in an untouched rain forest area in the Central Amazon, about 150 km northeast of the city of Manaus, Brazil. An overview of the atmospheric, geographic, and ecological conditions at the ATTO site has been published recently by Andreae et al. (2015). In this paper, a detailed description of the aerosol setup for the long-term measurements can be found. The instrumentation for CCN measurements is part of a broad aerosol measurement setup, which also covers aerosol size and concentration, absorptivity, scattering, fluorescence, as well as chemical composition (Andreae et al., 2015). The aerosol inlet is located at a height of 60 m, which is about 30 m above the forest canopy. The sample air is dried by silica gel diffusion dryers at the main inlet, which keeps the relative humidity (RH) below 40 %. For the CCN setup, a second diffusion dryer decreases the RH even further to < 20 %, which ensures reliable hygroscopicity measurements.

The CCN measurements are ongoing since the end of March 2014. This study covers the measurement period from end of March 2014 until February 2015, representing almost a full seasonal cycle. Also, the measurement period overlaps with the international large-scale field campaign GoAmazon2014/5 that was conducted in and around the city of Manaus from 1 January 2014 through 31 December 2015. During GoAmazon2014/5, comprehensive CCN measurements were conducted at different sites (see Sect. 1.3) (Martin et al., 2016). The ATTO site served as clean background T0a site during GoAmazon2014/5. Furthermore, the measurement period of this study overlaps with the German-Brazilian ACRIDICON-CHUVA field measurement campaign in





September 2014 (Machado et al., 2014; Wendisch et al., 2016), where (non-size-resolved) CCN measurements at multiple supersaturation levels were performed on board of the high altitude and long-range research aircraft (HALO) flying over the Amazon Basin.

### 2.2 Size-resolved CCN measurements

The number concentration of CCN was measured with a continuous-flow streamwise thermal gradient CCN counter (CCNC, model CCN-100, DMT, Boulder, CO, USA) (Roberts and Nenes, 2005; Rose et al., 2008b). The inlet flow rate of the CCNC was 0.5 L min$^{-1}$ with a sheath-to-aerosol flow ratio of 11. The water pump was operated at a rate of 4 mL h$^{-1}$ corresponding to the CCNC setting of "low" liquid flow. The supersaturation ($S$) of the CCNC was cycled through 10 different $S$ values between 0.11 % and 1.10 % (see Table 1), which are defined by controlled temperature gradients inside the CCNC column. Particles with a critical supersaturation ($S_c$) $\leq S$ in the column are activated and form water droplets. Droplets with diameters $\geq 1$ µm are detected by an optical particle counter (OPC) at the exit of the column.

Size-resolved *CCN activation curves* (for nomenclature see Sect. 2.3) were measured following the procedures in Rose et al. (2008a) and Krüger et al. (2014) by combining the CCNC with a differential mobility analyzer (DMA, model M, Grimm Aerosol Technik, Ainring, Germany). The DMA was operated with a sheath-to-aerosol flow ratio of 5. The DMA selects particles with a certain diameter ($D$) in the size range of 20 to 245 nm (sequence of $D$ value has been optimized for every $S$), which are then passed into the two instruments: (i) the CCNC system and (ii) a condensation particle counter (CPC, model 5412, Grimm Aerosol Technik), which measures the number concentration of aerosol particles with selected $D$ ($N_{CN}(D)$), while the CCNC measures the number concentration of CCN with selected $D$ for the given $S$ ($N_{CCN}(S,D)$). The cycle through a full CCN activation curve ($N_{CCN}(S,D)/N_{CN}(D)$) for one $S$ level took ~ 28 min, including ~ 40 s equilibration time for every new $D$, and ~ 2 min equilibration time for every new $S$ level. The completion of a full measurement cycle comprising CCN activation curves for 12-13 $D$ values (number of $D$ depends on $S$) and 10 different $S$ levels took ~ 4.5 h. The entire CCN system (including the CCNC, DMA, and CPC) was controlled by a dedicated LabView (National Instruments, München, Germany) routine.

The $S$ levels of the CCNC system were calibrated frequently (March, May, and September 2014) using ammonium sulfate ((NH$_4$)$_2$SO$_4$, Sigma Aldrich, St. Louis, MO, USA) particles, which were generated in an aerosol nebulizer (TSI Inc., Shoreview, MN, USA). The calibration procedure was conducted according to Rose et al. (2008b). All three calibrations gave consistent results and, thus, confirmed that the $S$ cycling in the CCNC was very stable and reliable throughout the entire measurement period.

All data presented here are given for ambient conditions. During the entire measurement period, no significant fluctuations in temperature (~28 °C) and pressure (~100 kPa) were observed in the air conditioned laboratory container.

### 2.3 Data analysis, error analysis, and nomenclature of CCN key parameters

The theoretical background and related CCN analysis procedures are comprehensively described elsewhere (Petters and Kreidenweis, 2007; Rose et al., 2008a). For the present study, the following corrections were applied to the data set: (i) The CCN activation curves were corrected for systematic deviations in the counting efficiency



of the CCNC and CPC according to Rose et al. (2010). (ii) Typically, the double-charge correction of the CCN activation curve is conducted according to Frank et al. (2006). For this study, we developed the following alternative approach, which *reconstructs* the CCN efficiency curves based on data from an independent scanning mobility particle sizer (SMPS, TSI model 3080 with CPC 3772 operating with standard TSI software) at the ATTO

site. The activation curve for every $D$ can be described by the following equation:

$$\frac{\sum_i N_{CCN}(S, D_i)}{\sum_i N_{CN}(D_i)} = \frac{\sum_i f(D_i) * s(D_i) * a(S, D_i)}{\sum_i f(D_i) * s(D_i)} \qquad (1)$$

The index $i$ represents the charge of the particles (typically $1 \leq i \leq 4$). The left side of the equation is the measured (non-corrected) ratio of CCN to CN for one selected $D$ and $S$. The parameter $s(D_i)$ is the multi-charge corrected particle number size distribution inverted from the SMPS measurements at $D_i$ with its different charge states. The

parameter $f(D_i)$ is the corresponding fraction of particles with the charge $i$. The function $a(S,D_i)$ accounts for the activated fraction of $s(D_i)$ at a given supersaturation $S$. We describe $a(S,D_i)$ as a cumulative Gaussian. Using a non-linear least square fit method (Levenberg-Marquardt) together with the knowledge of $s(D_i)$ and $f(D_i)$ the parameters of the function $a(S,D_i)$ can be optimized to get an optimal fit of the measured CCN activation curve for a given $S$. The function $a(S,D)$ is the cumulative Gaussian after the fit, which describes the multi-charge-corrected CCN

activation curve and has been used as a basis for the further analysis. Because the information of multiple charged particles also contributes to the fit results, this approach is superior to previously used methods, where this information is neglected. Based on $a(S,D)$, the critical diameter ($D_a(S)$, where 50 % of the particles are activated) is used to retrieve the effective hygroscopicity parameter ($\kappa(S,D_a)$) according to the $\kappa$-Köhler model (Petters and Kreidenweis, 2007). A detailed description of the calculation can be found in Petters and Kreidenweis (2007),

Rose et al. (2010), and Mikhailov et al. (2009).

The CCN size distribution ($N_{CCN}(S,D)$) was calculated by:

$$N_{CCN}(S, D) = s(D) * a(S, D) \qquad (2)$$

In this equation $s(D)$ represents the particle number size distribution of the SMPS at $D$ ($10 \leq D \leq 450$ nm).

The *CCN efficiencies* ($N_{CCN}(S)/N_{CN,10}$, for nomenclature see end of Sect. 2.3) have been calculated based on

the integral concentration of condensation nuclei (CN) with lower size cut-off $D_{cut} = 10$ nm ($N_{CN,10}$)[2] and CCN ($N_{CCN}(S)$) as:

$$\frac{N_{CCN}(S)}{N_{CN,10}} = \frac{\int_D N_{CCN}(S, D) * dD}{\int_D s(D) * dD} \qquad (3)$$

In addition to $D_a$, the maximum activated fraction ($MAF(S)$) can be obtained from $a(S,D)$. $MAF(S)$ typically equals unity, except for completely hydrophobic particles (i.e., fresh soot). The third parameter, which can be derived

from $a(S,D)$ is the width of the CCN activation curve $\sigma(S)$, which strongly depends on $D_a$. The ratio between $\sigma(S)$ and $D_a(S)$ ($\sigma(S)/D_a(S)$) is called heterogeneity parameter and can be used as an indicator for the chemical and the geometric diversity of the aerosol particles.

---

[2] Note that $N_{CN,10}$ corresponds to the total number size distribution for the characteristic size distribution at the ATTO site as there is a negligible aerosol population in the nucleation mode range (i.e., < 10 nm).





The error of $S$ was calculated based of the uncertainty according to the commonly used calibration procedure (Rose et al., 2008b). Overall, the error $\Delta S$ of $S$ equals approximately 10 %, however, in the following analysis we have used the specific $\Delta S$ values for every $S$ (see Table 1). The uncertainty of the selected $D$ of the DMA ($\Delta D$) was obtained as the mean width of the Gaussian fit of polystyrene latex (PSL) beads and equals 5.3 nm. For $N_{CCN}(S,D)$ and $N_{CN}(D)$ the standard error of the counting statistic was used. By Gaussian error propagation we determined $\Delta(N_{CCN}(S,D)/N_{CN}(D))$ and then repeated the data analysis for the upper and lower bounds $(1\pm\Delta)*(N_{CCN}(D,S)/N_{CN}(D))$. The resulting relative errors of the values $N_{CCN}(S)$, $N_{CN,10}$ and $N_{CCN}(S)/N_{CN,10}$ do not depend on $S$ and equal 6 %. The errors of $D_a$ and $\kappa(S, D_a)$ depend on $S$ and can be described as:

$$\Delta D_a = D_a * (S * 0.07 + 0.03) \qquad (4)$$

$$\Delta\kappa(S, Da) = \kappa(S, Da) * (S * 0.17 + 0.10) \qquad (5)$$

The use of certain terms in the context of CCN measurements is not uniform in the literature. For clarity, we summarize the key parameters and terms applied in this study as follows: (i) the value $N_{CCN}(S,D)/N_{CN}(D)$ is called *CCN activated fraction*, while (ii) $N_{CCN}(S,D)/N_{CN}(D)$ plotted against $D$ is called *CCN activation curve*; (iii) $N_{CCN}(S)$ plotted against $S$ is called *CCN spectrum*; (iv) $N_{CCN}(S)/N_{CN,Dcut}$ at a certain $S$ level is called *CCN efficiency*; (v) $N_{CCN}(S)/N_{CN,Dcut}$ plotted against $S$ is called *CCN efficiency spectrum*.

### 2.4 Aerosol mass spectrometry

In addition to the CCN measurements, aerosol chemical speciation monitor (ACSM, Aerodyne Research Inc., Billerica, MA, USA) measurements are being performed at the ATTO site (Andreae et al., 2015). The ACSM routinely characterizes non-refractory submicron aerosol species such as organics, nitrate, sulfate, ammonium, and chloride (Ng et al., 2011). Particles are focused by an aerodynamic lens system into a narrow particle beam, which is transmitted through three successive vacuum chambers. In the third chamber, the particle beam is directed into a hot tungsten oven (600 °C) where particles are flash-vaporized, ionized with a 70 eV electron impact ionizer, and detected with a quadrupole mass spectrometer. In this study, a time resolution of 30 minutes was used. The measurements provide a total mass concentration of the chemical composition of the aerosol particles. Further details about the ACSM can be found in (Ng et al., 2011).

### 2.5 Carbon monoxide measurements

Carbon monoxide (CO) measurements are conducted continuously at the ATTO site using a G1302 analyzer (Picarro Inc. Santa Clara, CA, USA). The experimental setup from the point of view of functioning and performance is a duplication of the system described in Winderlich et al. (2010).

### 3 Results and discussion

### 3.1 Time series of CCN parameters for the entire measurement period

Over the almost one-year measurement period from 25 March 2014 to 5 February 2015 we recorded size-resolved CCN activation curves at 10 different levels of water vapor supersaturation $S$ with an overall time resolution of approximately 4.5 hours. A total number of 10,253 CCN activation curves were fitted and analyzed to obtain parameters of CCN activity as detailed above (Sect. 2.3). Table 1 serves as a central reference in the course of this study and summarizes the annual mean values and standard deviations of the following key parameters, resolved



by $S$: $D_a(S)$, $\kappa(S,D_a)$, $\sigma(S)$, $\sigma(S)/D_a(S)$, $MAF(S)$, $N_{CCN}(S)$, $N_{CN,10}$, and $N_{CCN}(S)/N_{CN,10}$. In Fig. 1, some of these CCN key parameters are presented as time series over the entire measurement period to provide a general overview of their temporal evolution and variability. Concentration time series of the pollution tracers $N_{CN,10}$ and CO are added to illustrate the overall seasonality at the ATTO site.

Figure 1a displays the characteristic seasonal cycle in $N_{CN,10}$ and the CO mole fraction ($c_{CO}$). Both pollution tracers reach their maxima during the dry season ($N_{CN,10} = 1400 \pm 710$ cm$^{-3}$; $c_{CO} = 144 \pm 45$ ppb), whereas the smallest values are observed during the wet season ($N_{CN,10} = 285 \pm 131$ cm$^{-3}$; $c_{CO} = 117 \pm 12$ ppb) (given as mean ± one standard deviation). An obvious feature of the dry season months is the occurrence of rather short and strong peaks (reaching up to $N_{CN,10} = {\sim}5000$ cm$^{-3}$; $c_{CO} = {\sim}400$ ppb) on top of elevated background pollution levels.

The pronounced peaks originate from biomass burning plumes, which impact the ATTO site for comparably short periods (few hours up to several days). Selected events are discussed in detail in M. L. Pöhlker et al. (2016b). Figure 1b shows that $N_{CCN}(S)$ follows the same overall trends. A rather close correlation between $N_{CCN}(S)$ and $N_{CN,10}$ as well as $N_{CCN}(S)$ and $c_{CO}$ can be observed, as pointed out in previous studies (Andreae, 2009; Kuhn et al., 2010). Figure 1c displays the $\kappa(S,D_a)$ time series for three exemplary $S$ levels. It shows that the $\kappa(S,D_a)$ values,

which provide indirect information of the particles' chemical composition, are remarkably stable throughout the year (see also standard deviations of $\kappa(S,D_a)$ in Table 1). This illustrates that the dry season maximum in $N_{CCN}(S)$ is mainly related to the overall increase in $N_{CN,10}$, and not to substantial variations in aerosol composition and therefore $\kappa(S,D_a)$. Furthermore, this observation is consistent with the previously reported similarity between the CCN efficiency of Amazonian wet and dry season aerosol (Andreae et al., 2004). The levels of the three $\kappa(S,D_a)$

time series, with their corresponding $D_a$, provide a first indication that $\kappa(S,D_a)$ shows a clear size dependence, as further discussed in Sect. 3.2. The pronounced (but rather rare) 'spikes' in $\kappa(S,D_a)$ (i.e., in April and August) as well as various other specific events in this time series are analyzed in detail in the companion part 2 paper (M. L. Pöhlker et al., 2016b). Figure 1d gives an overview of the CCN efficiencies $N_{CCN}(S)/N_{CN,10}$ (for three $S$ levels) and its seasonal trends. This representation shows *continuously* high fractions of cloud-active particles for higher $S$

(e.g., $N_{CCN}(1.10\ \%)/N_{CN,10} > 0.9$) throughout the entire measurement period with almost no seasonality. For intermediate $S$ such as 0.47 %, the values of $N_{CCN}(0.47\ \%)/N_{CN,10}$ range from 0.6 to 0.9 and reveal a noticeable seasonal cycle, with highest levels during the dry season. Further, $N_{CCN}(0.11\ \%)/N_{CN,10}$ is mostly below 0.4 with clear seasonal trends. These observations can be explained by the characteristic aerosol size distribution at the ATTO site (Andreae et al., 2015), which (i) is dominated by particles in the Aitken (annually averaged peak $D_{Ait}$

at ~ 70 nm) and accumulation modes (annually averaged peak $D_{Acc}$ at ~ 150 nm), (ii) shows a sparse occurrence of nucleation mode particles (< 30 nm), and (iii) reveals a clear seasonality in the relative abundance of Aitken and accumulation modes (see Sect. 3.3 and Fig. 6). Thus, the higher dry season abundance of accumulation mode particles, which are more prone to act as CCN, result in higher $N_{CCN}(S)/N_{CN,10}$ levels, particularly at lower $S$.

  Analogous $N_{CCN}(S)/N_{CN}$ results from other continental background sites have been published previously: for

example, Levin et al. (2012) reported $N_{CCN}(0.97\%)/N_{CN} = 0.4\text{-}0.7$, $N_{CCN}(0.56\%)/N_{CN} = 0.25\text{-}0.5$, and $N_{CCN}(0.14\%)/N_{CN} < 0.15$ for a semi-arid Rocky Mountain site. Jurányi et al. (2011) reported $N_{CCN}(1.18\%)/N_{CN,16} = 0.6\text{-}0.9$, $N_{CCN}(0.47\%)/N_{CN,16} = 0.2\text{-}0.6$, and $N_{CCN}(0.12\%)/N_{CN,16} < 0.25$ for the high alpine Jungfraujoch site. At both locations, the CCN efficiencies tend to be lower than the corresponding results at the ATTO site, which can be explained by the frequent occurrence of new particle formation (NPF) and the related abundance of ultrafine





particles (with sizes well below $D_a(S)$) (Ortega et al., 2014; Boulon et al., 2010). The activated fractions at the Rocky Mountain and Jungfraujoch sites have a stronger seasonality than those at ATTO, probably inversely related to the seasonal cycle in NPF. Overall, we state that the activated fractions in the central Amazon, due the absence of significant ultrafine particle (<30 nm) populations, tend to be constantly higher than in other continental

background locations. The absence of 'classical' NPF (Kulmala et al., 2004) and corresponding lack of ultrafine particles is a unique property of the Amazon atmosphere resulting in the uniquely high CCN efficiencies.

The *MAF(S)* time series in Fig. 1e represents a valuable additional parameter to determine the abundance of 'poor' CCN (i.e., aerosol particles, which are not activated into CCN within the tested *S* range). For higher *S* (i.e., *S* > 0.11 %), *MAF(S)* is close to unity over the whole year. In contrast, *MAF(0.11 %)* fluctuates around unity during

the wet season months, however, it drops below unity during the biomass burning impacted dry season and subsequent transition period. For some episodes, *MAF(S)* shows very pronounced dips, as further outlined in the part 2 study (M. L. Pöhlker et al., 2016b).

**3.2 Annual means of CCN activation curves and hygroscopicity parameter**

Figure 2 displays the annual mean *CCN activation curves* for all *S* levels. Thus, it represents an overall characterization of the particle activation behavior, which means that for decreasing *S* levels the activation diameter, $D_a$, increases. In other words, every *S* corresponds to a certain (and to some extent typical) $D_a$ range, where particles start to become activated (see Table 1). As an example, relatively high *S* conditions (0.47-1.10 %) yield substantial activation already in the Aitken mode range, while low *S* levels (0.11-0.29 %) correspond to

activation of larger particles, mostly in the accumulation mode. Note that *S* levels in convective clouds rarely exceed 1.0 %, but that in the presence of precipitation higher *S* are possible (Cotton and Anthes, 1989). A close look reveals a gap between the activation curves for *S* = 0.47 % and *S* = 0.29 %, which corresponds to a jump in $\kappa(S,D_a)$ (discussed below). Moreover, the gap relates – in a way – to the bimodal size distribution and the characteristic Hoppel minimum (at 97 nm for the annual mean size distribution, see Table 2) between Aitken and

accumulation mode, as *S* = 0.47 % represents the onset of significant activation in the Aitken mode size range.

A different representation of these observations is displayed in Fig. 3, which shows the bimodally fitted (bimodal logarithmic normal distribution, $R^2 = 0.99$) annual mean $N_{CN}(D)$ size distribution. In this annual average representation, the Aitken mode maximum is located at $D_{Ait} = 69\pm1$ nm, the accumulation mode maximum at $D_{Acc} = 149\pm2$ nm, and both are separated by the Hoppel minimum (compare Table 2) (Hoppel et al., 1996).

Furthermore, Fig. 3 clearly shows that different $\kappa(S,D_a)$ values are retrieved for the Aitken ($\kappa_{Ait} = 0.14 \pm 0.03$) *versus* the accumulation mode size range ($\kappa_{Acc} = 0.22 \pm 0.03$). This indicates that Aiken and accumulation mode particles have different hygroscopicities and, thus, different chemical compositions. In this case, Aitken mode particles tend to be more predominantly organic (close to $\kappa = 0.1$) than the accumulation mode particles, which tend to contain more inorganic species (i.e., ammonium, sulfates, potassium etc.) (Gunthe et al., 2009; Prenni et

al., 2007; Wex et al., 2009; C. Pöhlker et al., 2012). The enhanced hygroscopicity in the accumulation mode is a well-documented observation for various locations worldwide, which is thought to result from the cloud processing history of this aerosol size fraction (e.g., Paramonov et al., 2013; Paramonov et al., 2015). For the Amazon Basin, our observed size dependence of $\kappa(S,D_a)$ agrees well with the values reported by Gunthe et al. (2009) and Whitehead et al. (2016).



The arithmetic mean hygroscopicity parameter at the ATTO site for all sizes (43 nm $< D_a <$ 172 nm) and for the entire measurement period is $\kappa_{mean} = 0.17 \pm 0.06$. For comparison, Gunthe et al. (2009) reported $\kappa_{mean} = 0.16 \pm 0.06$ (for the early wet season 2008). The observed standard deviation is rather small, which reflects the low variability of $\kappa_{mean}$ throughout the year (see Fig. 1b).

No perceptible diurnal trend in $\kappa_{mean}$ is present in the annually-averaged data. This is because the ATTO site is not (strongly) influenced by aerosol compositional changes that follow pronounced diurnal cycles (i.e., input of anthropogenic emissions). A consequence of this finding is that the overall hygroscopicity of the aerosol at the ATTO site (as a representative measurement station of the central Amazon) is well represented in model studies by using $\kappa_{mean} = 0.17 \pm 0.06$ (see also Sect. 3.5.4). Previous long-term CCN observations from alpine, semi-arid,
and boreal background sites have similarly shown that diurnal cycles in $\kappa(S,D_a)$ (or the related $D_a(S)$) tend to be rather small or even absent (Paramonov et al., 2013; Levin et al., 2012; Juranyi et al., 2011).

Figure 4, combines the annually averaged size distributions of $N_{CN}(D)$ as well as $N_{CCN}(S,D)$ for all $S$ levels. These curves result from multiplying the $N_{CN}(D)$ size distribution with the CCN activation curves in Fig. 2 and clearly visualize the inverse relationship of $D_a$ and $S$. Following the previous discussion of Fig. 2, $S$ ranging
between 0.11 % and 0.29 % mostly activates accumulation mode particles, while $S$ ranging between 0.47 % and 1.10 % activates the accumulation mode plus a substantial fraction of Aitken mode particles. For the highest supersaturation ($S = 1.10$ %) that was used in this study, almost the entire $N_{CN}(D)$ size distribution is being activated into CCN, which (regarding the very sparse occurrence of particles $< 30$ nm) explains the high $N_{CCN}(1.10\ \%)/N_{CN,10}$ levels in Fig. 1d.

### 3.3 Seasonal differences in CCN properties at the ATTO site

Within the seasonal periods in the central Amazon as defined in Sect. 1.2, we have subdivided the annual data set into the following four *periods of interest*, which represent the contrasting aerosol conditions and/or sources: (a) The first half of the wet seasons 2014 and 2015 received substantial amounts of long-range transport (LRT) aerosol,
mostly African dust, biomass smoke, and fossil fuel emissions (C. Pöhlker et al., 2016a; Ansmann et al., 2009; Salvador et al., 2016). Here, the corresponding period of interest will be called *LRT season* and covers 24 March to 13 April 2014 and 9 January to 10 February 2015; (b) In the late wet season 2014, all pollution indicators approached background conditions. Thus, the period 13 April to 31 May 2014 will be treated as clean *wet season* in this study. (c) The months June to July represent the transition period from wet to dry season and will be called
*transition wet to dry*. (d) The period of interest that covers the *dry season* with frequent intrusion of biomass burning smoke ranges from August to December 2014.

Figure 5 shows the CCN activation curves for all $S$ levels, subdivided into the four seasonal periods of interest. Although the plots for the individual seasons appear to differ only subtly, e.g., in $D_a(S)$ position and curve width, there is one major difference: the variable shape of the activation curve for the smallest $S = 0.11$ %. Particularly,
the behavior of *MAF(0.11%)* shows clear seasonal differences. It reaches unity during the *wet season*, whereas it levels off below unity for the *LRT*, *transition* and particularly for the *dry season* periods. The fraction of non-activated particles with $D \leq 245$ nm at $S = 0.11$ % is ~10 % during the *transition period* and ~20 % during the *dry season*. Interestingly, this effect is only observed for $S = 0.11$ %, whereas *MAF(>0.11 %)* reaches unity throughout the entire year. An explanation for this observation could be the intrusion of relatively fresh biomass burning





aerosol plumes during the *transition period* and *dry season*, which contain a fraction of comparably inefficient CCN. Soot is probably a main candidate here; however, fresh soot should also significantly reduce the *MAF(S)* for higher $S$ levels (Rose et al., 2010). Thus, we speculate that probably 'semi-aged' soot particles may be an explanation for the observed activation behavior.

Figure 6 corresponds to Fig. 3 and subdivides the annual mean $\kappa(S,D_a)$ size distribution ($\kappa(S,D_a)$ plotted against all measured $D_a(S)$) as well as the annual mean $N_{CN}(D)$ size distribution into their seasonal counterparts. The particle size distributions were fitted with a bimodal logarithmic normal distribution and the corresponding results are listed in detail in Table 2. The differences in the characteristic size distributions for the individual seasons clearly emerge: in addition to the strong variations in total particle number concentration (see Fig. 1), the

accumulation mode overwhelms the Aitken mode during the dry season, while accumulation and Aitken modes occur at comparable strength under wet season conditions. In other words, during the dry season, Aitken mode particles account on average for about 26 % in number of the total aerosol population ($N_{CN,Ait} = 483\pm49$ cm$^{-3}$ *versus* $N_{CN,Acc} = 1349\pm47$ cm$^{-3}$), whereas during the wet season, the Aitken mode accounts for about 62 % ($N_{CN,Ait} = 246\pm9$ cm$^{-3}$ *versus* $N_{CN,Acc} = 145\pm8$ cm$^{-3}$) (see Table 2). The size distribution of the *transition period*

from wet to dry season represents an intermediate state between the wet and dry season 'extremes'. Furthermore, the comparison between *wet season* condition *with* and *without LRT* influence reveals comparable distributions. However, a slight increase in the accumulation mode during LRT conditions indicates the presence of dust, smoke, pollution, and aged sea spray on top of the biogenic aerosol population during pristine periods (C. Pöhlker et al., 2016a).

The Hoppel minimum $D_H$ (Hoppel et al., 1996) between the Aitken and accumulation modes[3] also shows seasonal variations with its largest values around 110 nm in the wet season and its smallest values around 95 nm in the dry season (compare Fig. 5 and Table 2). Following Krüger et al. (2014) the observed $D_H$ can be used to determine an effective average cloud peak supersaturation $S_{cloud}(D_H,\kappa)$. Cloud development and dynamics are highly complex processes, in which aerosol particles are activated at different supersaturations. In the context of

this study, $S_{cloud}(D_H,\kappa)$ is used as a mean cloud supersaturation and serves as an overall reference value, however, it does not reflect the complex development of $S$ inside a cloud. Based on our data, $S_{cloud}(D_H,\kappa)$ is estimated as values around 0.29 % during dry season conditions and around 0.22 % during wet season conditions (Table 2). This indicates that $S_{cloud}(D_H,\kappa)$ levels tend to be noticeable lower during wet season cloud development compared to the dry season scenario. A plausible explanation for the comparably small $D_H$ and high $S_{cloud}(D_H,\kappa)$ in the dry

season could be the invigorated updraft regimes due to stronger solar heating. As outlined in Sect. 1.1, aerosol particle size, concentration, and hygroscopicity as well as cloud supersaturation represent key parameters for a detailed understanding of cloud properties. Fig. 6 provides reference values for all these parameters, resolved by seasons and thus provides a comprehensive insight into the Amazonian cloud properties.

---

[3] The position of $D_H$ was determined as the intersection of the fitted and *normalized* modes (monomodal fits for Aitken and accumulation mode were normalized to equal area). The normalization is necessary for a precise localization of $D_H$ because large difference in Aitken and accumulation mode strength (e.g., for the dry season conditions) cause biased $D_H$ as the intersection of both modes is shifted towards the smaller mode.



Comparing the seasonal $\kappa(S,D_a)$ size distributions in Fig. 6, it is obvious that the (seasonally averaged) $\kappa_{Ait}$ values in the Aitken mode size range are surprisingly stable between 0.13 and 0.14 throughout the whole year. This indicates that the Aitken mode aerosol population was persistently dominated by almost pure organic particles throughout the seasons. In contrast, noticeable seasonal differences were observed for (seasonally averaged) $\kappa_{Acc}$

values in the accumulation mode size range, with mean values ranging from around 0.21 to 0.28. This indicates that the accumulation mode also comprises high contents of organic materials, however with elevated amounts of inorganic ingredients (i.e., sulfate, ammonium, and potassium). In the size range around $D_H$, which separates the (apparently) chemically distinct aerosol populations of Aitken and accumulation modes, a step-like increase in $\kappa(S,D_a)$ is observed. The highest seasonally averaged $\kappa(S,D_a)$ values (up to 0.28) are observed during intrusion of

dust, marine sulfate, and sea salt-rich LRT plumes. Note that short-term peaks in $\kappa(S,D_a)$ can be even higher (see case studies in part 2 paper (M. L. Pöhlker et al., 2016b)). In the absence of LRT, the $\kappa_{Acc}$ values are also rather stable for most of the year and range between 0.21 and 0.24. Overall, a remarkable observation is the high similarity between the wet and dry season $\kappa(S,D_a)$ size distributions, while many other aerosol parameters undergo *substantial* seasonal variations (Andreae et al., 2015).

The $\kappa(S,D_a)$ levels reported here agree well with the corresponding results in the previous Amazonian CCN studies by Gunthe et al. (2009) and Whitehead et al. (2016), which range between 0.1 and 0.4, with a mean around 0.16±0.06. In a wider context, our results also agree well with previous long-term measurements at other continental background locations (i.e., alpine, semi-arid, and boreal sites) (Paramonov et al., 2013; Levin et al., 2012; Jurányi et al., 2011). Comparing the four sites with each other the following observations can be made: (i)

$\kappa_{Ait}$ tends to be smaller than $\kappa_{Acc}$ at all four background locations. (ii) At the alpine, semi-arid, and boreal sites, $\kappa(S,D_a)$ undergoes a rather *gradual* increase from the Aitken to the accumulation mode size range (Paramonov et al., 2013 and references therein), whereas this increase appears to be *steeper* (step-like) in the Amazon. This can clearly be seen in the present study (e.g., Fig. 3) as well as in Gunthe et al. (2009) and Whitehead et al. (2016). (iii) Particularly in the vegetated environments (i.e., tropical, boreal, and semi-arid forests), $\kappa_{Ait}$ mostly ranges

between 0.1 and 0.2, suggesting that the Aitken mode particles predominantly comprise organic constituents. Furthermore, $\kappa_{Ait}$ shows a remarkably small seasonality for these locations. (iv) The $\kappa_{Acc}$ levels show a much wider variability throughout the seasons for all locations.

Figure 7 presents the diurnal cycles in $\kappa_{mean}$ for the four seasonal periods of interest. No perceptible diurnal trends in $\kappa_{mean}$ can be observed for any of the seasons. The only observable difference is an increased variability

of $\kappa_{mean}$ during the LRT season (see error bars in Fig. 7a). This can be explained by the episodic character of LRT intrusions, which causes an 'alternating pattern' of clean periods with background conditions and periods of elevated concentrations of LRT aerosol (C. Pöhlker et al., 2016a).

### 3.4 Aerosol chemical composition and effective hygroscopicity

Continuous ACSM measurements are being conducted at the ATTO site since March 2014, providing online and non-size resolved information on the chemical composition of the non-refractory aerosol (Andreae et al., 2015). Here, we compare the ACSM data on the aerosol's chemical composition with the CCNC-derived $\kappa(S,D_a)$ values.



This analysis focusses on the dry season months, when ACSM and CCNC were operated in parallel.[4] Note that the ACSM covers a size range from 75 nm to 650 nm (Ng et al., 2010), while the size resolved CCN measurements provide information up to particle sizes of about 170 nm. Since the ACSM records the size-integrated masses of defined chemical species (organics, nitrate, sulfate, ammonium, and chloride), the results tend to be dominated by the fraction of larger particles with comparably high masses (i.e., in the accumulation mode size range) and influenced less by the fraction of small particles with comparably low masses (i.e., in the Aitken mode size range). Thus, in order to increase the comparability between ACSM and CCNC, we have chosen the lowest $S$ level ($S = 0.11\pm0.01$ %), which represents the largest measured $D_a$ ($D_a = 172\pm12$ nm).

In Fig. 8, the $\kappa(0.11\%,D_a)$ values are plotted against the ACSM-derived organic mass fraction ($f_{org}$). The data was fitted with (i) a linear fit and (ii) a bivariate regression according to Cantrell (2008). A linear fit approach was used by Gunthe et al. (2009) to determine the effective hygroscopicity parameters $\kappa_{org} = 0.1$ of biogenic Amazonian SOA ($f_{org} = 1$) and $\kappa_{inorg} = 0.6$ for the inorganic fraction ($f_{org} = 0$). For the present data set, the same procedure results in an acceptable coefficient of determination ($R^2 = 0.66$). We estimated the effective hygroscopicity parameters $\kappa_{org}=0.12\pm0.01$ and $\kappa_{inorg}=0.61\pm0.01$ based on the linear fit and extrapolation to $f_{org} = 1$ and $f_{org} = 0$, respectively. This is in good agreement with previous studies (Gunthe et al., 2009; Rose et al., 2011; King et al., 2007; Engelhart et al., 2008). However, a drawback of the linear fitting approach is the fact that swapping $f_{org}$ and $\kappa(0.11\%,D_a)$ on the axes will change the results.

Therefore, we also applied the bivariate regression fit, which takes into account that both parameter, $f_{org}$ and $\kappa(0.11\%,D_a)$, have an experimental error. For the bivariate regression an error of 5% in $f_{org}$ and an error of 10% in $\kappa(0.11\%,D_a)$ were used. A coefficient of determination of $R^2 = 0.71$ was obtained for the bivariate regression, which is slightly better than for the linear fit. Based on the bivariate regression, we estimated effective hygroscopicity parameters $\kappa_{org}=0.10\pm0.01$ and $\kappa_{inorg}=0.71\pm0.01$ for the organic and inorganic fractions, respectively.

### 3.5 CCN parametrizations and prediction of CCN number concentrations

Cloud-resolving models on all scales – spanning from large eddy simulations (LES) to global climate models (GCM) – require simple and efficient parametrizations of the complex microphysical basis to adequately reflect the spatiotemporal CCN cycling (Cohard et al., 1998; Andreae, 2009). Previously, several different approaches to predict CCN concentrations have been suggested (Rose et al., 2010; Deng et al., 2013; Gunthe et al., 2009; Andreae, 2009). Any parametrization strategy seeks an efficient combination of a minimal set of input data, on one hand, and a good representation of the atmospheric CCN population, on the other hand.

The detailed analysis in this study has shown that the CCN population in the central Amazon is mainly defined by comparably stable $\kappa$ levels, due to the predominance of organic aerosol particles, and rather pronounced seasonal trends in aerosol number size distribution. Particularly, the remarkably stable $\kappa(S,D_a)$ values suggest that the Amazonian CCN cycling can be parametrized rather precisely for efficient prediction of CCN concentrations.

---

[4] Although the ACSM measurement has been started in March 2014, instrumental issues during the initial months cause some uncertainty for the corresponding data. Thus, for this study we focus only on the data period (Aug to Dec 2014), when the instrumental issues were resolved.





In the following paragraphs, we apply the following CCN parametrization strategies to the present data set and explore their strengths and limitations:

    (i)      CCN prediction based on the correlation between $N_{CCN}(0.4\%)$ and $N_{CN}$, called here the $\Delta N_{CCN}(0.4\%)/\Delta N_{CN}$ *parametrization*,

(ii)     CCN prediction based on the correlation between $N_{CCN}(S)$ and $c_{CO}$, called here the $\Delta N_{CCN}(S)/\Delta c_{CO}$ *parametrization*,

    (iii)    CCN prediction based on analytical fit functions of experimentally obtained CCN spectra, called *CCN spectra parametrization*,

    (iv)    CCN prediction based on the *κ*-Köhler model, called *κ-Köhler parametrization*, and

(v)     CCN prediction based on a novel and effective parametrization built on CCN efficiency spectra, called *CCN efficiency spectra parametrization*.

The prediction accuracy for the individual strategies is summarized in Table 3.

### 3.5.1 $\Delta N_{CCN}(0.4\%)/\Delta N_{CN}$ parametrization

Andreae (2009) analyzed CCN data sets from several contrasting field sites worldwide and found significant relationships between the satellite-retrieved aerosol optical thickness (AOT) and the corresponding $N_{CCN}(0.4\%)$ levels as well as between the total aerosol number concentration $N_{CN}$ and $N_{CCN}(0.4\%)$. The obtained ratio $N_{CCN}(0.4\%)/N_{CN} = 0.36\pm0.14$ – in other words the globally averaged CCN efficiency at $S = 0.4$ % – can be used to predict CCN concentrations. The corresponding results for the present data set are displayed in Fig. 9a and show a surprisingly tight correlation, given that a globally obtained $N_{CCN}(0.4\%)/N_{CN}$ ratio has been used. However, Fig. 9a also shows a systematic underestimation of the predicted CCN concentration $N_{CCN,p}(0.4\%)$, which can be explained by the comparably high activated fractions in the Amazon (e.g., $N_{CCN}(0.47\%)/N_{CN,10}$ ranging from 0.6 to 0.9; see Fig. 1). Activated fractions in other locations worldwide tend to be lower due to the (more persistent) abundance of nucleation mode particles, as discussed in Sect. 3.1.

In Sec. 3.5.5 we will show that our novel parametrization is an extension of this approach: The $N_{CCN}(0.4\%)/N_{CN}$ *parametrization* refers to a globally averaged CCN efficiency at one specific $S$, while the *CCN efficiency spectra parametrization* is based on an analytical description of CCN efficiencies across the entire (relevant) $S$ range and has been determined specifically for the central Amazon.

### 3.5.2 $\Delta N_{CCN}(S)/\Delta c_{CO}$ parametrization

Experimentally obtained excess $N_{CCN}(S)$ to excess $c_{CO}$ ratios can be used to calculate $N_{CCN,p}(S)$. Kuhn et al. (2010) determined $\Delta N_{CCN}(0.6\%)/\Delta c_{CO} = \sim 26$ cm$^{-3}$ ppb$^{-1}$ for biomass burning plumes and $\Delta N_{CCN}(0.6\%)/\Delta c_{CO} = \sim 49$ cm$^{-3}$ ppb$^{-1}$ for urban emissions in the area around Manaus, Brazil. Lawson et al. (2015) have investigated biomass burning emissions in Australia and found $\Delta N_{CCN}(0.5\%)/\Delta c_{CO} = 9.4$ cm$^{-3}$ ppb$^{-1}$. In the context of the present study,

we have calculated $\Delta N_{CCN}(S)/\Delta c_{CO}$ for a strong biomass burning event in August 2014. This event and its impact on the CCN population is subject of a detailed discussion in the companion part 2 paper (M. L. Pöhlker et al., 2016b). Here, we use the $\Delta N_{CCN}(S)/\Delta c_{CO}$ ratios from the companion paper to obtain a CCN prediction. The observed $\Delta N_{CCN}(S)/\Delta c_{CO}$ ratios range between $6.7\pm0.5$ cm$^{-3}$ ppb$^{-1}$ (for $S = 0.11$ %) and values around $18.0\pm1.3$ cm$^{-3}$ ppb$^{-1}$ (for higher $S$) (see summary in Table 4). Since biomass burning is the dominant source of pollution in the



central Amazon, these biomass burning-related $\Delta N_{CCN}(S)/\Delta c_{CO}$ ratios in Table 4 have been used to calculate $N_{CCN,p}(S)$ for the present data set. The corresponding results in Fig. 9b show a reasonable correlation for highly polluted conditions ($N_{CN} > 2000$ cm$^{-3}$) and a poor correlation for cleaner states ($N_{CN} < 2000$ cm$^{-3}$). This behavior can be explained by the fact that the high concentrations in CCN and CO originate from frequent biomass burning

plumes during the Amazonian dry season (see Fig. 1). Thus, they can be assigned to the same sources with rather defined $\Delta N_{CCN}(S)/\Delta c_{CO}$ ratios (Andreae et al., 2012). During the contrasting cleaner periods, CN and CO originate from a variety of different sources, which are often not related and, therefore, explain the poor correlation for clean to semi-polluted conditions. Overall, Fig. 9b indicates that the quality of CO-based CCN prediction is rather poor, due to the complex interplay of different sources. The overall deviation between $N_{CCN,p}(S)$ and $N_{CCN}(S)$ for this

approach is about 170 % (Table 3).

### 3.5.3 Classical and improved CCN spectra parametrization

The total number of particles that are activated at a given $S$ is regarded as one of the central parameters in cloud formation and evolution (Andreae and Rosenfeld, 2008). Thus, *CCN spectra* ($N_{CCN}(S)$ plotted against $S$) are a

widely and frequently used representation in various studies to summarize the observed $N_{CCN}(S)$ values over the cloud-relevant $S$ range for a given time period and location (Martins et al., 2009b; Gunthe et al., 2009; Twomey and Wojciechowski, 1969; Roberts et al., 2002; Rissler et al., 2004; Freud et al., 2008). Different analytical fit functions of the experimental CCN spectra have been proposed and are used as parametrization schemes for $N_{CCN}(S)$ in modelling studies (e.g., Deng et al., 2013; Khain et al., 2000; Pinsky et al., 2012; Cohard et al., 1998).

In the context of the present study, the annual mean Amazonian CCN spectrum is shown in Fig. 10. As an analytical representation of the experimental data, we have used Twomey's empirically found (classical) power law fit function (Twomey, 1959)

$$N_{CCN}(S) = N_{CCN}(1\%) * \left(\frac{S}{1\%}\right)^k \quad (6)$$

which yields a reasonable coefficient of determination of $R^2 = 0.88$ (Fig. 10a). The obtained fit parameters

$N_{CCN}(1\%) = 998$ cm$^{-3}$ (sometimes also called $c$) and $k = 0.36$ agree with results from previous measurements that are summarized by Martins et al. (2009b). The power law function has become a widely used parametrization due to its simplicity (Cohard et al., 1998). However, it is based on strong assumptions as well as not related to the physical basis of the fitted data and thus reveals certain drawbacks, such as the poor representation of $N_{CCN}(S)$ at small $S$ (i.e., < 0.2 %) as well as the fact that for larger $S$ (i.e., > 1.2 %) it does not converge against $N_{CN}$ which is,

for physical reasons, the upper limit.

As an alternative, an error function fit – which is used in this context for the first time – represents the data much better (Fig. 10b). The proposed error function (erf)

$$N_{CCN}(S) = A * \text{erf}\left(\frac{ln\left(\frac{S}{S_0}\right)}{width_0}\right) \qquad (7)$$

is related to the physical basis of the fitted data and yields a high coefficient of determination $R^2 = 0.997$.

Mathematically, this erf represents an integration of a log-normal $N_{CN}(D)$ size distribution. Analogously, the $N_{CCN}(D)$ spectrum represents the *cumulative* distribution of the *relative* $N_{CN}(D)$ distribution (compare Fig. 4). A double-erf fit would be even more appropriate for the bimodal Amazon $N_{CN}(D)$ distribution (compare Fig. 6 and



discussion in Sect. 3.5.5). However, the single-erf fit proposed above proved to be (already) a very good analytical representation as underlined by the high coefficient of determination ($R^2 > 0.99$). The erf fit reflects the physically expected saturation behavior of aerosol activation for high $S$ and, thus, converges against a limit of $A = 1067 \pm 22$ cm$^{-3}$, which matches well with the mean total number concentration $N_{CN,10} = 1097 \pm 66$ cm$^{-3}$. The erf fit (if not forced

through the origin) transects the abscissa at $S_0 = 0.066$ %. Therefore, the erf fit cannot describe the CCN activation behavior for low $S$ ($\leq 0.07$ %), which is also an experimentally not accessible $S$ range.

Figure 11a and b show the corresponding $N_{CCN,p}(S)$ *versus* $N_{CCN}(S)$ scatter plots.[5] In general, parametrizations based on CCN spectra yield a mean state based on average concentrations (see fit parameters in Fig. 10) and ignore the temporal variability of the aerosol concentrations (Rose et al., 2010; Martins et al., 2009a; Jurányi et al., 2011).

On closer inspection, Table 3 shows that the erf fit allows somewhat better predictions (deviation of power law fit about 227 % *versus* 215 % for erf fit), which can be explained by the fact that the erf fit presents the experimental data more appropriately (compare Fig. 10). Overall, however, the power law fit and the erf fit approaches give rather poor correlations, due to the missing representation of the aerosol's temporal variability, which is an inherent limitation of the CCN spectra parametrization. It can be concluded that this parametrization requires a minimum

of aerosol input data (i.e., only the parameters of the corresponding fit function), which explains its wide distribution in various modelling attempts. However, Fig. 10 and Table 3 show that this simplicity is clearly at the expense of the prediction accuracy.

### 3.5.4 $\kappa$-Köhler parametrization

The $\kappa$-Köhler model approach has been used in previous studies and gave good CCN predictions (e.g., Gunthe et al., 2009; Rose et al., 2010). For the present data set, the $N_{CCN,p}(S)$ concentrations were calculated according to Rose et al. (2010).[6] Here, the annually averaged values $\kappa_{Ait} = 0.14$ and $\kappa_{Acc} = 0.22$ were used for the CCN prediction, since they accurately represent the stable $\kappa$ levels in the central Amazon. Figure 12 shows the corresponding $N_{CCN,p}(S)$ *versus* $N_{CCN}(S)$ scatter plot, in which the areas with highest density of data points precisely

follow the one-to-one line. Table 3 underlines this good agreement as the observed deviation of around 10 % between $N_{CCN,p}(S)$ and $N_{CCN}(S)$ is the smallest among all tested parametrizations. Accordingly, the $\kappa$-Köhler model approach turns out to be a very accurate parametrization. However, it requires a time series of $N_{CN}$ size distributions as input data and is therefore the most 'data demanding' strategy in this regard.

### 30    3.5.5 CCN efficiency spectra parametrization

It has to be kept in mind that CCN spectra strongly depend on the total aerosol concentration and, thus, predominantly reflect the specific (temporary) aerosol population during the period of the study. The shape of CCN spectra provides some information on the aerosol activation behavior as a function of $S$. However, the strong variability in the total aerosol abundance makes it difficult to compare the CCN *efficiency* behavior between

---

[5] The horizontal lines in the scatter plots results from the fact that constant $N_{CCN,p}(S)$ values are obtained for the different $S$ levels.

[6] Briefly, for every SMPS scan the $N_{CN}$ size distribution has been integrated above the critical diameter $D_a$, in which $D_a$ has been obtained based on a given $\kappa$ and $S$.



different locations and/or periods of interest with specific (e.g., seasonal) conditions. For the present dataset, Fig. 13 shows annually averaged *CCN efficiency spectra* ($N_{CCN}(S)/N_{CN,Dcut}$ plotted against $S$) for two different reference aerosol concentrations $N_{CN,10}$ and $N_{CN,50}$.[7] The corresponding fit parameters are summarized in Table 5. The CCN efficiency spectra are independent of the total aerosol load and, instead, reflect the fraction of activated particles for the relevant $S$ range. Here, we also use an erf fit

$$\frac{N_{CCN}(S)}{N_{CN,Dcut}} = \frac{1}{2} + \frac{1}{2} * \mathrm{erf}\left(\frac{ln\left(\frac{S}{S_1}\right)}{width_1}\right) \qquad (8)$$

to describe the data, for the same reasons as outlined in Sect. 3.5.3. The fits yield high coefficients of determination ($R^2 = 0.99$). Per definition, $N_{CCN}(S)/N_{CN,Dcut}$ spans from zero to unity. Therefore, the offset $y_0$ of the function as well as the pre-factor $A$ have been set to 0.5. For the atmospherically relevant $S$ range – typically $S < 0.6\%$, see Andreae (2009) – aerosol sizes around 50-60 nm are considered as the onset of the CCN size range (see also Fig. 4). Accordingly, if $D_{cut}$ is chosen close to this activation threshold, the corresponding $N_{CCN}(S)/N_{CN,Dcut}$ approaches unity, which can be seen in Fig. 13. The free variable $S_1$ (e.g., $S_1 = 0.22\pm0.01\%$ for $N_{CN,10}$ and $S_1 = 0.19\pm0.01\%$ for $N_{CN,50}$) represents the $S$ value where half of the aerosol particles are activated into cloud droplets. A monodisperse aerosol with a defined composition would yield a steep step-like *CCN efficiency spectrum*, while the complex Amazonian aerosol results in a wide and rather smooth 'step'. In other words, the width of the erf fit (here $width_1 = 1.78\pm0.08$ for $N_{CN,10}$ and $width_1 = 1.41 \pm0.05$ for $N_{CN,50}$) is an (indirect) measure for the diversity (i.e., size and composition) of the aerosol population.

Figure 14 shows a direct comparison of the CCN efficiency spectra resolved by seasonal periods of interest (compare also Sect. 3.3), which reveals characteristic differences in the curve's shape (i.e., its 'steepness'). The corresponding fit parameters are summarized in Table 5. A good numeric indicator for the differences in 'steepness' is the fit parameter $S_1$, which specifies the 50 % activation supersaturation of the total aerosol population. The largest contrast in shape and $S_1$ can be seen between the dry and wet season scenario: During the dry season the CCN efficiency increases steeply with $S$, and $S_1$ is reached at 0.18 % for $N_{CN,10}$, whereas during the wet season, the increase of the CCN efficiency is rather 'modest' and $S_1$ is reached only at 0.35 % for $N_{CN,10}$. The transition period represents (once more) an intermediate state between the dry and wet season 'extremes' ($S_1 = 0.28\%$ for $N_{CN,10}$). For transition period conditions, Kuhn et al. (2010) reported $N_{CCN}(0.6\%)/N_{CN} = 0.66 \pm 0.15$, which is in good agreement with Fig. 14c ($N_{CCN}(0.61\%)/N_{CN,10} = 0.72 \pm 0.10$).

The observed differences among the CCN efficiency spectra in Fig. 14 reflect some of the major trends in the aerosol seasonality in Amazonia. A closer look at Fig. 6 helps to understand those. Overall, the key parameters in the CCN activation behavior are (primarily) the aerosol number size distribution and, in a secondary role, the particles' chemical composition, represented by $\kappa(S,D_a)$ (Dusek et al., 2006). Thus, the seasonally averaged number size distributions and the seasonally averaged $\kappa(S,D_a)$ size distribution in Fig. 6 have to be considered to explain the different shapes in Fig. 14. Focusing on the contrasting wet and dry season plots it can be stated that: (i) While

---

[7] The use of aerosol number concentrations with $D_{cut} = 50$ nm has been suggested by Paramonov et al. (2015) as a reference value to ensure comparability of CCN efficiencies from different studies.



the $\kappa(S,D_a)$ size distribution for wet and dry season appear to be very similar (same size trend and same values), the number size distributions (i.e., the ratio of Aitken and accumulation modes) differ substantially. (ii) With increasing $S$ the diameter $D_a$ decreases and is shifted from the accumulation towards the Aitken mode size range. (iii) Thus, under dry season conditions, comparably small $S$ levels ($S = 0.11\text{-}0.2$ %) can already activate most

particles of the pronounced accumulation mode. (iv) In contrast, under wet season conditions, while the same $S$ levels still activate the accumulation mode particles, the comparably strong Aitken mode remains unactivated. This means that the ratio of Aitken and accumulation mode particles ($N_{CN,Ait}/N_{CN,Acc}(\text{wet}) = 1.7$; $N_{CN,Ait}/N_{CN,Acc}(\text{dry}) = 0.4$; compare Table 2) determines the activated fraction as a function of $S$ and, thus, also the steepness of the CCN efficiency spectra in Fig. 14.

While size appears as the dominant parameter in the CCN activation behavior, in certain cases variability in chemical composition also matters (Dusek et al., 2006). In Fig. 14, this can be seen for the wet season cases *with* and *without* LRT influence: In the presence of LRT aerosol the 50% activation occurs already at $S_1 = 0.22$ % for $N_{CN,10}$, which is much closer to the dry ($S_1 = 0.18$ % for $N_{CN,10}$) than to the wet season ($S_1 = 0.35$ % for $N_{CN,10}$) behavior. While Fig. 6 shows that the number size distributions for both cases are similar, the observed difference

in Fig. 14 can be explained by the deviations in the corresponding $\kappa(S,D_a)$ size distributions. In other words, the elevated $\kappa(S,D_a)$ levels during the intrusion of LRT aerosols allows the activation of particle sizes that remain inactivated at the lower $\kappa(S,D_a)$ levels in the absence of LRT aerosol. Therefore, the differences in chemical composition can explain the decreased $S_1$ in these cases.

In Fig. 14, *single-erf* fits have been used as analytical descriptions of the CCN efficiency spectra. Overall, this

approach provides a good representation of the experimental data (see high coefficients of determination in Table 5). However, the single erf fit is merely an approximation, assuming that the aerosol size distribution is monomodal. This is a valid assumption for the dry season (see Fig. 6) and corresponds with a good agreement between fit and data points in Fig. 14d. In contrast, the wet season shows pronounced and prevailing bimodal size distributions (see Fig. 6), which corresponds to a clear discrepancy between the fit and data points in Fig. 14b (i.e.,

for S > 0.3 %). For a bimodal size distribution, a *double-erf* fit is the physically more appropriate description (see also discussion in Sect. 3.5.3). Figure 15 illustrates the contrast between a single and a double-erf fit of the wet season CCN efficiency spectrum for $N_{CN,50}$. As expected, the double-erf fit is clearly a better representation of the data across the entire $S$ range. However, in the context of this study, the double-erf fit of CCN spectra merely serves as proof of concept. It will be discussed in more detail in a follow-up study (M. L. Pöhlker et al., 2016c).

Thus, in the context of the following CCN parametrization, we will work exclusively with the single-erf fit approach for the following reasons: (i) the single-erf fit represents the simpler parametrization scheme (2 fit parameters instead of 6) and (ii) the difference in the CCN prediction accuracy of single *versus* double-erf fit turns out to be insignificant.

Figure 16 explores the applicability of the *CCN efficiency spectra parametrization* (single-erf fits) to calculate

CCN concentrations. The following four modifications of the parametrization scheme are compared: *annually average* CCN efficiency spectra with (i) $D_{cut} = 10$ nm and (ii) $D_{cut} = 50$ nm (compare Fig. 13) as well as *seasonally resolved* CCN efficiency spectra with (iii) $D_{cut} = 10$ nm, and (iv) $D_{cut} = 50$ nm (compare Fig. 14). All cases in Fig. 16 show rather tight correlations, which prove the high prediction accuracy of the CCN efficiency spectra parametrization. The corresponding deviations between $N_{CCN}(S)$ and $N_{CCN,p}(S)$ are summarized in Table 3. The



comparison confirms that the cases with $D_{cut} = 50$ nm perform better than $D_{cut} = 10$ nm. Moreover, the seasonally resolved cases show higher prediction accuracies than the annually averaged scenarios. Thus, the highest deviation of 33 % is observed for case Fig. 16a and the lowest deviation (and therefore best performance) with 17 % for case Fig. 16d (see Table 3).

5     In a way, the CCN efficiency spectra parametrization represent a 'compromise' between the previously introduced parametrization strategies: It operates with a comparably small set of input data and still provides good prediction accuracies. The input data requires the fit parameters $S_1$ and $width_1$ of the single-erf fit, which reflects the 'shape' of the fit functions. This part conveys the specific CCN activation behavior of the given aerosol population (e.g., the wet season scenario). Moreover, a time series of $N_{CN,Dcut}$ is required, which accounts for the

10    temporal variability of the aerosol population. The new parametrization approach is currently extended and applied to further datasets worldwide (M. L. Pöhlker et al., 2016c).





### 4 Conclusions

Size-resolved CCN measurements have been conducted at the remote ATTO site in the central Amazon, spanning a full seasonal cycle from March 2014 until February 2015. These measurements represent the first long-term study on CCN concentrations and hygroscopicity in this unique and globally important ecosystem. The reported measurements span the aerosol size range of 20 - 245 nm and, therefore, cover the Aitken and accumulation modes, which dominate the aerosol burden in the Amazon throughout the year (Andreae et al., 2015). The supersaturation in the CCN counter was cycled through 10 levels from $S = 0.11$ % to $S = 1.10$ %. Overall, this study presents an in-depth analysis of the key CCN parameters, based on a continuous sequence of more than 10,000 CCN activation curves with a temporal resolution of 4.5 h and, therefore, allows a detailed analysis of the CCN cycling in the central Amazon Basin.

The Amazonian atmosphere reveals a characteristic bimodal aerosol size distribution, which is dominated by pronounced Aitken and accumulation modes ($D_{Ait} \sim 70$ nm *versus* $D_{Acc} \sim 150$ nm) as well as the sparse occurrence of nucleation mode particles ($< 30$ nm). This size distribution closely relates to the observed CCN properties, as its entire size range – and thus the majority of particles – falls into the CCN-active range. Accumulation mode particles are CCN-active at supersaturations between 0.11 and 0.29 %, while supersaturations between 0.47 and 1.10 % activate both, the Aitken and accumulation modes. The absence of nucleation mode particles further explains the high activated fractions $N_{CCN}(S)/N_{CN,10}$ that were observed throughout all seasons, with $N_{CCN}(0.11$ %$)/N_{CN,10}$ reaching up to 0.4 and $N_{CCN}(1.1$ %$)/N_{CN,10}$ constantly exceeding 0.9. These values are substantially higher than corresponding activated fractions at other continental background sites worldwide (Levin et al., 2012; Jurányi et al., 2011; Paramonov et al., 2013). Overall, the CCN concentrations $N_{CCN}(S)$ for all $S$ levels closely follow the pronounced pollution-related seasonal cycle in $N_{CN}$ that is typical for the Amazon region.

The hygroscopicity parameter $\kappa(S,D_a)$, which reflects the chemical composition of the particles, appears to be remarkably stable throughout the entire measurement period with only a weak seasonal cycle and no perceptible diurnal trends. Numerically, the $\kappa(S,D_a)$ values lie within a rather narrow range from 0.1 to 0.3 for most of the time. The mean hygroscopicity averaged over the entire period and size range and its corresponding standard deviation is $\kappa_{mean} = 0.17 \pm 0.06$. In terms of particle size, $\kappa(S,D_a)$ reveals a clear size dependence with lower values for the Aitken mode ($\kappa_{Ait} = 0.14 \pm 0.03$) and elevated levels in the accumulation mode range ($\kappa_{Acc} = 0.22 \pm 0.05$). Previous studies showed that the Amazonian aerosol population is dominated by organic aerosols throughout the seasons (Chen et al., 2015; Martin et al., 2010b; Gunthe et al., 2009). The comparably low $\kappa(S,D_a)$ values in this study underline this observation. However, the observed difference between $\kappa_{Ait}$ and $\kappa_{Acc}$ shows that the Aitken mode is almost purely organic (close to $\kappa = 0.1$), while the accumulation mode is somewhat enriched in inorganic constituents.

Focusing on seasonal differences, substantial changes in the aerosol concentrations and the shape of the size distribution have been observed. During the (clean) wet season, equally strong Aitken and accumulation modes were observed, while during the (polluted) dry season the accumulation mode overwhelms the Aitken mode. The transition periods represent intermediate states between these extremes. Interestingly, the strong seasonal variability in aerosol abundance and sources does not correspond to noticeable changes in $\kappa(S,D_a)$. In other words, $\kappa_{Ait}$ and $\kappa_{Acc}$ are almost identical for dry and wet season conditions. The only seasonal period where $\kappa(S,D_a)$ deviates from its typical range is the LRT season when out-of-Basin dust, marine sulfate, and sea salt is transported into the





Amazon Basin. During this period, a significant increase in $\kappa_{Acc}$ up to 0.28 is observed. In summary, the seasonally averaged CCN populations (represented by the CCN efficiency spectra) are mostly defined by particle size (i.e., shape of aerosol size distribution). The only episodes when (besides size) chemical variability also matters are the LRT periods with their enhanced $\kappa(S,D_a)$ values.

5    Based on the CCN key parameters that have been extracted in the present study, we show that the CCN population over Amazonia could be modeled very effectively. Different approaches to infer a CCN concentration from basic aerosol parameters have been compared and it turns out that a remarkably good correlation between modelled and measured data can be obtained based on continuous SMPS time series as well as the annually averaged $\kappa_{Ait}$ and $\kappa_{acc}$ values from this study. Alternatively, CCN concentration can effectively be calculated based

10   on our novel parametrization, which is based on fitted CCN efficiency spectra and continuous time series of total aerosol number concentrations. These efficient approaches to infer the Amazonian CCN population will probably help to improve future modelling studies.





**Acknowledgements**

This work has been supported by the Max Planck Society (MPG) and the Max Planck Graduate Center with the Johannes Gutenberg University Mainz (MPGC). For the operation of the ATTO site, we acknowledge the support by the German Federal Ministry of Education and Research (BMBF contract 01LB1001A) and the Brazilian

Ministério da Ciência, Tecnologia e Inovação (MCTI/FINEP contract 01.11.01248.00) as well as the Amazon State University (UEA), FAPEAM, LBA/INPA and SDS/CEUC/RDS-Uatumã. This paper contains results of research conducted under the Technical/Scientific Cooperation Agreement between the National Institute for Amazonian Research, the State University of Amazonas, and the Max-Planck-Gesellschaft e.V.; the opinions expressed are the entire responsibility of the authors and not of the participating institutions. We highly

acknowledge the support by the Instituto Nacional de Pesquisas da Amazônia (INPA). We would like to especially thank all the people involved in the technical, logistical, and scientific support of the ATTO project, in particular Matthias Sörgel, Thomas Disper, Andrew Crozier, Uwe Schulz, Steffen Schmidt, Antonio Ocimar Manzi, Alcides Camargo Ribeiro, Hermes Braga Xavier, Elton Mendes da Silva, Nagib Alberto de Castro Souza, Adir Vasconcelos Brandão, Amauri Rodriguês Perreira, Antonio Huxley Melo Nascimento, Thiago de Lima Xavier, Josué Ferreira

de Souza, Roberta Pereira de Souza, Bruno Takeshi, and Wallace Rabelo Costa. Further, we thank the GoAmazon2014/5 team for the fruitful collaboration and discussions. We acknowledge technical support by the DMT and Grimm Aerosol Technik teams in the course of the experiments. Moreover, we thank Qiaoqiao Wang, Bettina Weber, Nina Ruckteschler, Bruna Amorim Holanda, Eugene Mikhailov, Kathrin Reinmuth-Selzle, J. Alex Huffman, Ramon Braga, and Daniel Rosenfeld for support and stimulating discussions.





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



**Table A1. List of used symbols.**

| Symbol | Quantity and Unit |
|---|---|
| $A$ | CN number concentration derived from erf fit of CCN spectra, cm$^{-3}$ |
| $a(S,D_i)$ | cumulative Gaussian fit of multi-charge CCN activation fraction at a given $D$ and $S$ |
| $a(S,D)$ | cumulative Gaussian fit of CCN activation fraction at a given $S$ |
| $c_{CO}$ | CO mole fraction, ppb |
| $D$ | mobility equivalent particle diameter, nm |
| $D_a(S)$ | midpoint activation diameter determined from CCN activation curve, nm |
| $D_{Ait}$ | position of Aitken mode maximum, nm |
| $D_{Acc}$ | position of accumulation mode maximum, nm |
| $D_{cut}$ | lower cut-off diameter in aerosol number reference concentration $N_{CN,Dcut}$, nm |
| $D_H$ | position of Hoppel minimum, nm |
| $f(D_i)$ | multiple-charged fraction at a given $D$ |
| $f_{org}$ | organic mass fraction |
| $f_{inorg}$ | inorganic mass fraction |
| $I$ | number of charges |
| $\kappa$ | hygroscopicity parameter |
| $\kappa(S,D_a)$ | hygroscopicity parameter determined from CCN activation curve |
| $\kappa_{Acc}$ | mean hygroscopicity parameter for accumulation mode particles |
| $\kappa_{Ait}$ | mean hygroscopicity parameter for Aitken mode particles |
| $\kappa_{mean}$ | mean hygroscopicity parameter for all measured $S$ |
| $MAF(S)$ | maximum activated fraction determined by CCN activation curve |
| $N$ | number of data points |
| $N_{CCN}(S)$ | CCN number concentration at a certain $S$, cm$^{-3}$ |
| $N_{CCN,p}(S)$ | predicted CCN number concentration at a certain $S$, cm$^{-3}$ |
| $N_{CCN}(S,D_a)$ | CCN number concentration determined from CCN activation curve, cm$^{-3}$ |
| $N_{CCN}(S,D) / N_{CN}(D)$ | CCN activation fraction |
| $N_{CCN}(S) / N_{CN,Dcut}$ | CCN efficiency for aerosol reference concentration $N_{CN,Dcut}$ |
| $N_{CN,Dcut}$ | aerosol number reference concentration ($>D_{cut}$), cm$^{-3}$ |
| $N_{CN,10}$ | aerosol number reference concentration ($>10$ nm), cm$^{-3}$ |
| $N_{CN,50}$ | aerosol number reference concentration ($>50$ nm), cm$^{-3}$ |
| $N_{CN,Acc}$ | CN number concentration for accumulation mode particles, cm$^{-3}$ |
| $N_{CN,Ait}$ | CN number concentration for Aitken mode paticles, cm$^{-3}$ |
| $S$ | water vapor supersaturation, % |
| $S_c$ | critical supersaturation for CCN activation, % |
| $S_{cloud}(D_H,\kappa)$ | average cloud peak supersaturation, % |
| $s(D)$ | SMPS size distribution, cm$^{-3}$ |
| $s(D_i)$ | multi charge size distribution of $D$, cm$^{-3}$ |
| $S_0$ | abscissa transect of erf fit of CCN spectra, % |
| $S_1$ | midpoint activation supersaturation determined from CCN efficiency spectra, % |
| $width_0$ | width of erf fit of CCN spectra |
| $width_1$ | width of erf fit of CCN efficiency spectra |
| $x_0$ | position of mobility equivalent particle diameter, nm |
| $\sigma$ | width of log-normal fit of Aitken and accumulation modes |
| $\sigma(S)$ | width of CCN activation curve, nm |
| $\sigma(S)/D_a(S)$ | heterogeneity parameter |



**Table A2. List of used acronyms.**

| Acronym | Description |
|---|---|
| ACSM | aerosol chemical speciation monitor |
| AOT | aerosol optical thickness |
| ATTO | Amazon tall tower observatory |
| ACRIDICON | aerosol, cloud, precipitation, and radiation interactions and dynamics of convective cloud systems |
| BUNIAACIC | Brazil-UK network for investigation of Amazonian atmospheric composition and impacts on climate |
| BC | black carbon |
| CCN | cloud condensation nuclei |
| CCNC | cloud condensation nuclei counter |
| CN | condensation nuclei |
| CHUVA | cloud processes of the main precipitation systems in Brazil: a contribution to cloud resolving modeling and to the GPM (global precipitation measurements) |
| CPC | condensation particle counter |
| CO | carbon monoxide |
| DMA | differential mobility analyzer |
| Erf | lognorm error function |
| GCM | global climate models |
| GoAmazon14/5 | green ocean Amazon 2014/5 |
| HALO | high altitude and long-range research aircraft |
| HTDMA | hygroscopicity tandem differential mobility analyzer |
| IN | ice nuclei |
| IOP | intensive observation period |
| LES | large eddy simulation |
| LRT | long-range transport |
| NPF | new particle formation |
| OPC | optical particle counter |
| PSL | polystyrene latex |
| RH | relative humidity |
| SE | standard error |
| SMPS | scanning mobility particle sizer |
| SOA | secondary organic aerosol |
| UTC | coordinated universal time |




**Table 1.** Characteristic CCN parameters as a function of the supersaturation $S$, averaged over the entire measurement period: midpoint activation diameter $D_a(S)$, hygroscopicity parameter $\kappa(S,D_a)$, width of CCN activation curve $\sigma(S)$, heterogeneity parameter $\sigma(S)/D_a(S)$, maximum activated fraction $MAF(S)$, CCN number concentration $N_{CCN}(S)$, total particle concentration ($> 10$ nm) $N_{CN,10}$, CCN efficiencies $N_{CCN}(S)/N_{CN,10}$, and number of data points $n$. $S$ is shown as set value $\pm$ the experimentally derived deviation in $S$. All other values are given as arithmetic mean $\pm$ one standard deviation. All values are provided for ambient conditions (temperature ~28 °C; pressure ~100 kPa).

| $S$ [%] | $D_a(S)$ [nm] | $\kappa(S,D_a)$ | $\sigma(S)$ [nm] | $\sigma(S)/D_a(S)$ | $MAF(S)$ | $N_{CCN}(S)$ [cm$^{-3}$] | $N_{CN,10}$ [cm$^{-3}$] | $N_{CCN}(S)/N_{CN,10}$ | $n$ |
|---|---|---|---|---|---|---|---|---|---|
| 0.11±0.01 | 172±12 | 0.22±0.05 | 45±11 | 0.26±0.06 | 0.93±0.10 | 275±219 | 1100±776 | 0.24±0.10 | 1071 |
| 0.15±0.02 | 136±10 | 0.22±0.05 | 42±10 | 0.31±0.06 | 0.97±0.05 | 457±384 | 1093±770 | 0.39±0.13 | 1086 |
| 0.20±0.02 | 117±9 | 0.21±0.05 | 35±10 | 0.30±0.07 | 0.98±0.04 | 571±482 | 1096±775 | 0.48±0.15 | 1087 |
| 0.24±0.03 | 105±8 | 0.19±0.05 | 29±8 | 0.28±0.07 | 0.99±0.04 | 652±550 | 1098±778 | 0.55±0.16 | 1078 |
| 0.29±0.03 | 98±7 | 0.17±0.04 | 27±8 | 0.27±0.08 | 1.01±0.05 | 719±601 | 1103±784 | 0.60±0.17 | 1069 |
| 0.47±0.04 | 77±5 | 0.13±0.03 | 17±6 | 0.22±0.07 | 1.03±0.04 | 883±744 | 1101±799 | 0.74±0.18 | 1008 |
| 0.61±0.06 | 63±4 | 0.14±0.03 | 15±5 | 0.23±0.07 | 0.97±0.03 | 900±719 | 1089±791 | 0.78±0.14 | 922 |
| 0.74±0.08 | 57±4 | 0.13±0.03 | 14±6 | 0.24±0.09 | 0.96±0.03 | 941±730 | 1108±809 | 0.82±0.12 | 984 |
| 0.92±0.11 | 49±4 | 0.13±0.03 | 12±6 | 0.24±0.11 | 0.96±0.04 | 987±742 | 1117±814 | 0.86±0.10 | 995 |
| 1.10±0.08 | 43±3 | 0.13±0.03 | 11±5 | 0.25±0.10 | 0.95±0.03 | 1013±747 | 1120±792 | 0.88±0.08 | 952 |





**Table 2.** Properties (position $x_0$, integral number concentration $N_{CN}$, width $\sigma$) of Aitken and accumulation modes from the double log-normal fit (compare $R^2$) of the total particle size distributions. Values are given as annual mean and subdivided into seasonal periods of interest as specified in Sect. 3.3. Compare also Fig. 6. In addition, values for position of Hoppel minimum $D_H$ as well as estimated average peak supersaturation in cloud $S_{cloud}(D_H,\kappa)$ are listed. The errors represent the uncertainty of the fit parameters. The error in $S_{cloud}(D_H,\kappa)$ is the experimentally derived error in $S$.

| season | Mode | $N_{CN}$ [cm$^{-3}$] | $\kappa$ | $x_0$ [nm] | $\sigma$ | $R^2$ | $D_H$ [nm] | $S_{cloud}(D_H,\kappa)$ [%] |
|---|---|---|---|---|---|---|---|---|
| year | Aitken | 397±31 | 0.13±0.03 | 69±1 | 0.44±0.02 | 0.99 | 97±2 | 0.29±0.03 |
| | accumulation | 906±29 | 0.22±0.05 | 149±2 | 0.57±0.01 | | | |
| LRT | Aitken | 231±8 | 0.14±0.04 | 67±1 | 0.63±0.01 | 0.99 | 109±2 | 0.23±0.02 |
| | accumulation | 232±10 | 0.28±0.08 | 172±1 | 0.51±0.01 | | | |
| wet | Aitken | 246±9 | 0.13±0.02 | 70±1 | 0.53±0.01 | 0.99 | 112±2 | 0.22±0.02 |
| | accumulation | 145±8 | 0.21±0.05 | 170±2 | 0.42±0.01 | | | |
| transition | Aitken | 405±24 | 0.14±0.02 | 65±1 | 0.42±0.01 | 0.99 | 92±2 | 0.34±0.03 |
| | accumulation | 668±24 | 0.24±0.04 | 135±1 | 0.53±0.01 | | | |
| dry | Aitken | 483±49 | 0.13±0.03 | 71±2 | 0.42±0.03 | 0.99 | 97±2 | 0.29±0.03 |
| | accumulation | 1349±47 | 0.21±0.04 | 150±2 | 0.58±0.01 | | | |



**Table 3.** Characteristic deviation between observed and predicted CCN number concentrations – $N_{CCN}(S)$ and $N_{CCN,p}(S)$ – based on different parametrization schemes, according to Rose et al. (2008). For every parametrization scheme and resolved by $S$ the following information is provided: (i) arithmetic mean values of the relative bias $\Delta_{bias}N_{CCN}(S) = (N_{CCN,p}(S)-N_{CCN}(S)) / N_{CCN}(S)$ and (ii) of the total relative deviation $\Delta_{dev}N_{CCN}(S) = |N_{CCN,p}(S)-N_{CCN}(S)| / N_{CCN}(S)$.

| S [%] | $\Delta N_{CCN}(S)/\Delta N_{CN}$ | | $\Delta N_{CCN}(S)/\Delta c_{CO}$ | | fits of CCN spectra | | | | κ-Köhler | | erf fit of CCN efficiency spectra | | | | | | | |
| | | | | | Twomey power law fit | | erf fit | | | | annual average | | | | resolved by seasons | | | |
| | | | | | | | | | | | $N_{CN,10}$ | | $N_{CN,50}$ | | $N_{CN,10}$ | | $N_{CN,50}$ | |
| | Bias | Dev | bias | dev | bias | dev | bias | dev | bias | dev | bias | dev | bias | dev | bias | dev | bias | dev |
|---|---|---|---|---|---|---|---|---|---|---|---|---|---|---|---|---|---|---|
| 0.11±0.01 | - | - | 1.48 | 1.75 | 4.68 | 4.75 | 2.54 | 2.81 | 0.18 | 0.22 | 0.64 | 0.74 | 0.24 | 0.44 | 0.39 | 0.53 | 0.14 | 0.36 |
| 0.15±0.02 | - | - | 0.50 | 1.21 | 2.78 | 2.99 | 2.42 | 2.69 | 0.07 | 0.11 | 0.27 | 0.47 | 0.10 | 0.32 | 0.15 | 0.36 | 0.04 | 0.27 |
| 0.20±0.02 | - | - | 2.84 | 2.96 | 2.46 | 2.75 | 2.60 | 2.86 | 0.11 | 0.13 | 0.22 | 0.43 | 0.13 | 0.30 | 0.14 | 0.33 | 0.08 | 0.24 |
| 0.24±0.03 | - | - | 1.78 | 1.98 | 1.93 | 2.26 | 2.24 | 2.50 | 0.09 | 0.10 | 0.16 | 0.37 | 0.12 | 0.25 | 0.12 | 0.28 | 0.09 | 0.20 |
| 0.29±0.03 | - | - | 2.19 | 2.33 | 1.74 | 2.09 | 2.12 | 2.39 | 0.14 | 0.14 | 0.22 | 0.42 | 0.14 | 0.25 | 0.17 | 0.32 | 0.11 | 0.20 |
| 0.40 | -0.41 | 0.47 | - | - | - | - | - | - | - | - | - | - | - | - | - | - | - | - |
| 0.47±0.04 | - | - | 1.33 | 1.54 | 1.36 | 1.73 | 1.70 | 1.93 | 0.04 | 0.06 | 0.09 | 0.26 | 0.07 | 0.16 | 0.08 | 0.20 | 0.06 | 0.12 |
| 0.61±0.06 | - | - | 1.02 | 1.15 | 1.23 | 1.55 | 1.47 | 1.73 | 0.08 | 0.09 | 0.08 | 0.18 | 0.05 | 0.09 | 0.08 | 0.15 | 0.05 | 0.08 |
| 0.74±0.08 | - | - | 1.50 | 1.59 | 1.22 | 1.51 | 1.37 | 1.63 | 0.09 | 0.10 | 0.09 | 0.16 | 0.04 | 0.06 | 0.09 | 0.14 | 0.04 | 0.06 |
| 0.92±0.11 | - | - | 1.11 | 1.28 | 1.15 | 1.42 | 1.18 | 1.44 | 0.08 | 0.08 | 0.05 | 0.10 | 0.01 | 0.03 | 0.05 | 0.09 | 0.01 | 0.04 |
| 1.10±0.08 | - | - | 1.12 | 1.25 | 1.11 | 1.35 | 1.05 | 1.31 | 0.08 | 0.08 | 0.04 | 0.08 | -0.01 | 0.04 | 0.05 | 0.08 | -0.01 | 0.05 |
| **All** | - | - | **1.50** | **1.73** | **2.00** | **2.27** | **1.89** | **2.15** | **0.10** | **0.11** | **0.19** | **0.33** | **0.10** | **0.20** | **0.14** | **0.25** | **0.06** | **0.17** |





**Table 4. Excess $N_{CCN}(S)$ to excess $c_{CO}$ ratios $\Delta N_{CCN}(S)/\Delta c_{CO}$ for the individual $S$ levels during peak period of the strong biomass burning event in August 2014. This event of analyzed in detail through a case study in the companion part 2 paper (M. L. Pöhlker et al., 2016b). The values $\Delta N_{CCN}(S)/\Delta c_{CO}$ were obtained from bivariate regression fit of scatter plots between $N_{CCN}(S)$ and $c_{CO}$ for individual $S$ levels (Andreae et al. 2012).**

| $S$ | $\Delta N_{CCN}(S) / \Delta CO$ | $N$ | $R^2$ |
|---|---|---|---|
| [%] | [cm$^{-3}$ ppb$^{-1}$] | [cm$^{-3}$] | |
| 0.11±0.01 | 6.7±0.5 | -603 ±125 | 0.86 |
| 0.15±0.02 | 13.6±1.4 | -1447 ±354 | 0.68 |
| 0.20±0.02 | 14.3±0.8 | -1128 ±208 | 0.90 |
| 0.24±0.03 | 16.8±1.0 | -1460 ±261 | 0.86 |
| 0.29±0.03 | 17.4±1.3 | -1378 ±296 | 0.83 |
| 0.47±0.04 | 20.1±1.7 | -1675 ±425 | 0.84 |
| 0.61±0.06 | 17.9±1.3 | -1206 ±332 | 0.88 |
| 0.74±0.08 | 16.5±1.3 | -933 ±329 | 0.88 |
| 0.92±0.11 | 18.1±1.4 | -1265 ±355 | 0.85 |
| 1.10±0.08 | 17.5±1.3 | -1096 ±328 | 0.87 |




**Table 5. Erf fit parameters describing CCN efficiency spectra $N_{CCN}(S)/N_{CN,Dcut}$ versus $S$ as model input data (compare Figs. 13 and 14). Fit parameters are provided for (i) annually averaged efficiency spectra with five different aerosol number references concentrations $N_{CN,Dcut}$ and (ii) resolved by seasons for $N_{CN,10}$ and $N_{CN,50}$.**

| $N_{CN,Dcut}$ | time period | $S_1$ [%] | $width_1$ | $R^2$ |
|---|---|---|---|---|
| $N_{CN,10}$ | | 0.22±0.01 | 1.78 ±0.08 | 0.99 |
| $N_{CN,20}$ | annual | 0.22±0.01 | 1.78 ±0.08 | 0.99 |
| $N_{CN,30}$ | | 0.22±0.01 | 1.72 ±0.07 | 0.99 |
| $N_{CN,50}$ | | 0.19±0.01 | 1.41 ±0.05 | 0.99 |
| $N_{CN,10}$ | wet season | 0.35±0.01 | 1.80 ±0.06 | 0.99 |
| | LRT period | 0.22±0.01 | 2.39±0.10 | 0.98 |
| | transition | 0.28±0.01 | 1.70 ±0.05 | 0.99 |
| | dry season | 0.18±0.01 | 1.57 ±0.11 | 0.98 |
| $N_{CN,50}$ | wet season | 0.26±0.01 | 1.37 ±0.12 | 0.99 |
| | LRT period | 0.17±0.01 | 1.58 ±0.10 | 0.99 |
| | transition | 0.23±0.01 | 1.38 ±0.04 | 0.99 |
| | dry season | 0.17±0.01 | 1.31 ±0.06 | 0.92 |




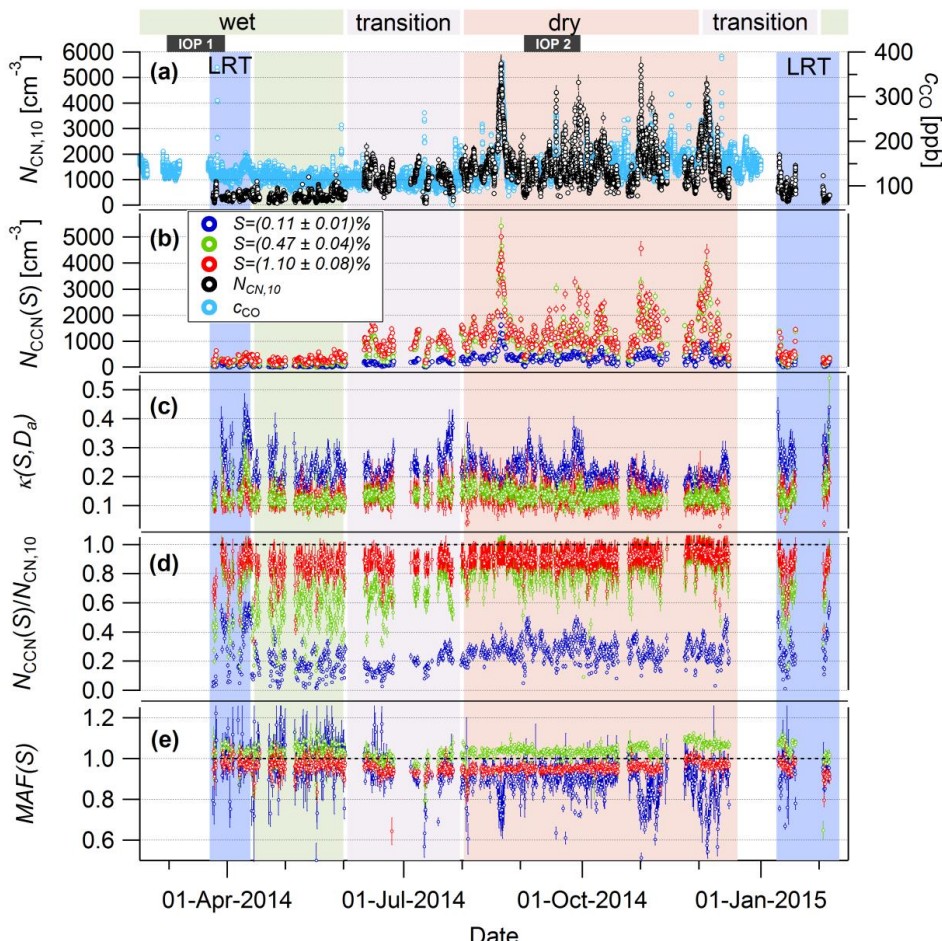

**Figure 1.** Seasonal trends in time series of total aerosol concentration $N_{CN,10}$, carbon monoxide mole fraction ($c_{CO}$), and CCN key parameters for three selected supersaturations $S$ for entire measurement period (shown in original time resolution). (a) Time series of pollution tracers $N_{CN,10}$ and $c_{CO}$. (b) CCN concentrations $N_{CCN}(S)$, (c) hygroscopicity parameter $\kappa(S,D_a)$, (d) CCN efficiencies $N_{CCN}(S)/N_{CN,10}$, and (e) maximum activated fraction $MAF(S)$. Three different types of shading represent: (i) the seasonality in the Amazon atmosphere according to Andreae et al. (2015) (wet *versus* dry seasons with transition periods, illustrated in top of graph), (ii) periods of IOP1 and IOP2 during GoAmazon2014/5, (iii) seasonal periods of interest in context of the present study as defined in Sect. 3.3 (shading in background of time series).





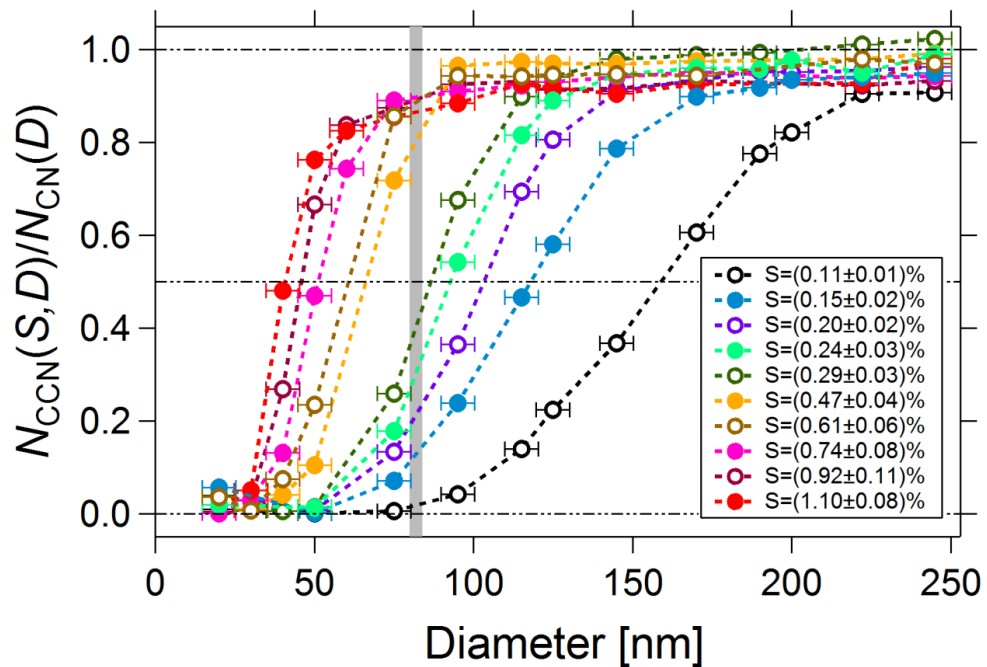

**Figure 2. CCN activation curves for all measured $S$ levels ($S$ = 0.11-1.10 %), averaged over entire measurement period. Data points represent arithmetic mean values. For $N_{CCN}(S,D)/N_{CN}(D)$ the standard error is plotted, which is very small (due to the large number of scans with comparably small variability) and, therefore, not perceptible in this representation. For the diameter $D$ the error bars represent the experimental error as specified in Sect. 2.3. The grey vertical band represents the position of the Hoppel minimum (including error range) for the annual mean number size distribution (compare Fig. 3). Dashed lines provide visual orientation and indicate 0, 50, and 100 % activation. The value at 50 % activation is used for calculation of the hygroscopicity parameter $\kappa(S,D_a)$. The lines connecting the data points merely serve as visual orientation.**





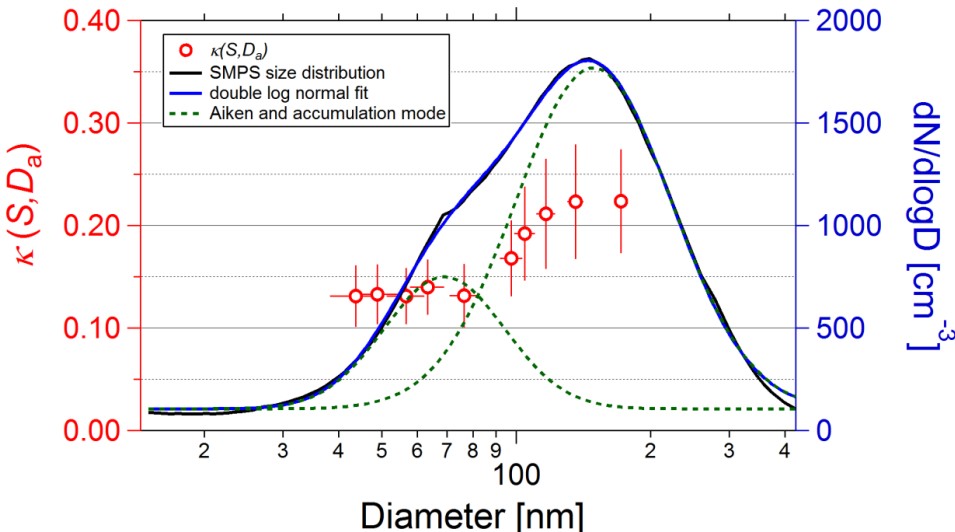

**Figure 3. Size dependence of hygroscopicity parameter $\kappa(S,D_a)$ averaged over entire measurement period. Values of $\kappa(S,D_a)$ for every $S$ level are plotted against their corresponding midpoint activation diameter $D_a$ (left axis). For $\kappa(S,D_a)$ the error bars represent one standard deviation. For $D_a$ the experimentally derived error is shown. In addition, the average number size distribution for the entire measurement period is shown (right axis). Dashed green lines represent the average Aitken and accumulation modes. The standard error of the number size distribution is indicated as grey shading, which is very small and therefore hardly perceptible in this representation due to the large number of scans with comparably small variability. Distinctly different $\kappa(S,D_a)$ levels can be observed for Aitken and accumulation modes with lower variability in Aitken than in accumulation mode.**





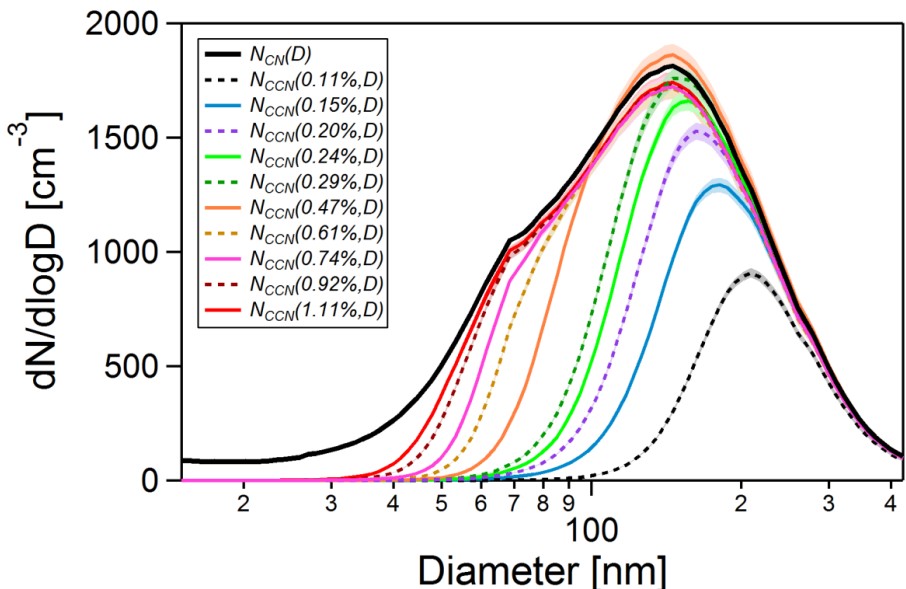

**Figure 4.** Number size distributions of total aerosol particles $N_{CN}(D)$ and of cloud condensation nuclei $N_{CCN}(S,D)$ at all 10 supersaturation levels ($S$ = 0.11-1.10 %) averaged over the entire measurement period. The $N_{CCN}(S,D)$ size distributions were calculated by multiplying the average $N_{CN}(D)$ size distributions (in Fig. 3) with average CCN activation curves in (Fig. 2).




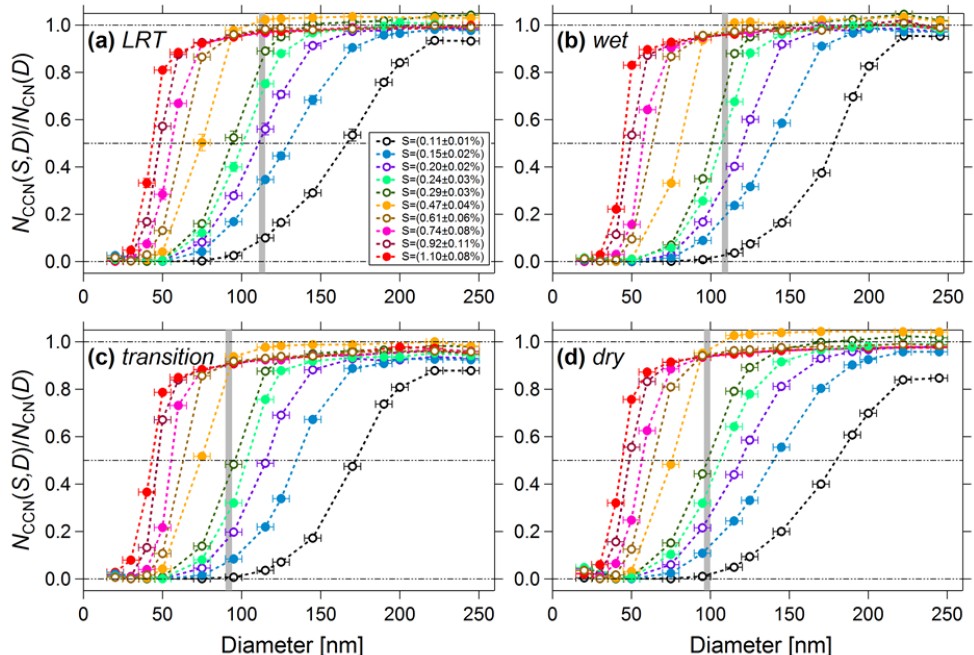

**Figure 5. CCN activation curves for all measured $S$ levels ($S$ = 0.11-1.10 %), subdivided into seasonal periods of interest as specified in Sect. 3.3. Data points represent arithmetic mean values. For $N_{CCN}(S,D)/N_{CN}(D)$ the standard error is plotted, which is very small (due to the large number of scans with comparably small variability) and, therefore, not perceptible in this representation. For the diameter $D$ the error bars represent the experimental error as specified in Sect. 2.3. The grey vertical bands represent the (seasonal) position of the Hoppel minima (including error range, compare Table 2). Dashed horizontal lines provide visual orientation and indicate 0, 50, and 100 % activation. The 50 % activation diameter is used for calculation of the hygroscopicity parameter $\kappa(S,D_a)$. The lines connecting the data points merely serve as visual orientation.**




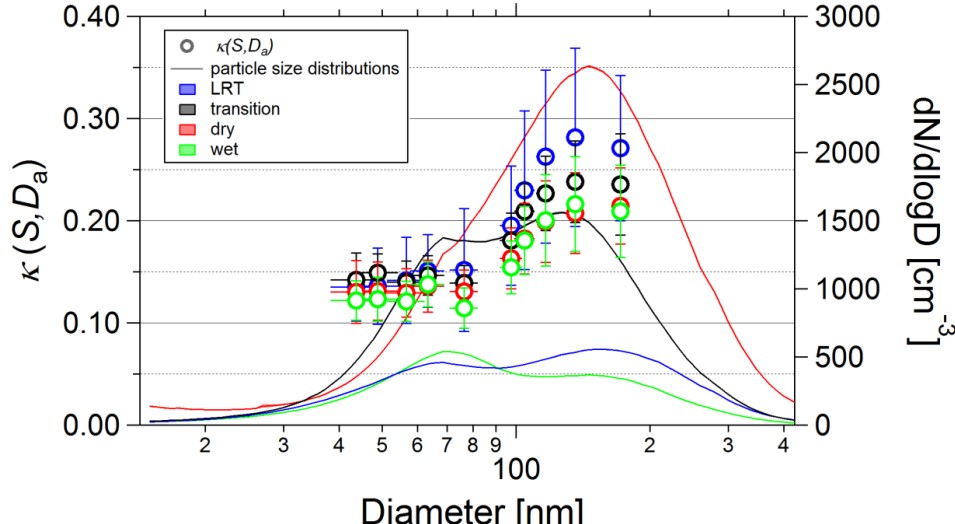

**Figure 6.** Size dependence of hygroscopicity parameter $\kappa(S,D_a)$ subdivided into seasonal periods of interest (color coding) as specified in Sect. 3.3. Values of $\kappa(S,D_a)$ for every $S$ level are plotted against their corresponding midpoint activation diameter $D_a$ (left axis). For $\kappa(S,D_a)$ the error bars represent one standard deviation. For $D_a$ the experimentally derived error is shown. In addition, the average number size distribution for the seasonal periods of interest are shown (right axis). The standard error of the number size distributions is indicated as colored shading, which is very small and therefore hardly perceptible in this representation due to the large number of scans with comparably small variability. A clear size dependence and seasonal trends in $\kappa(S,D_a)$ levels can be observed. The averaged number size distributions show very pronounced seasonal differences.




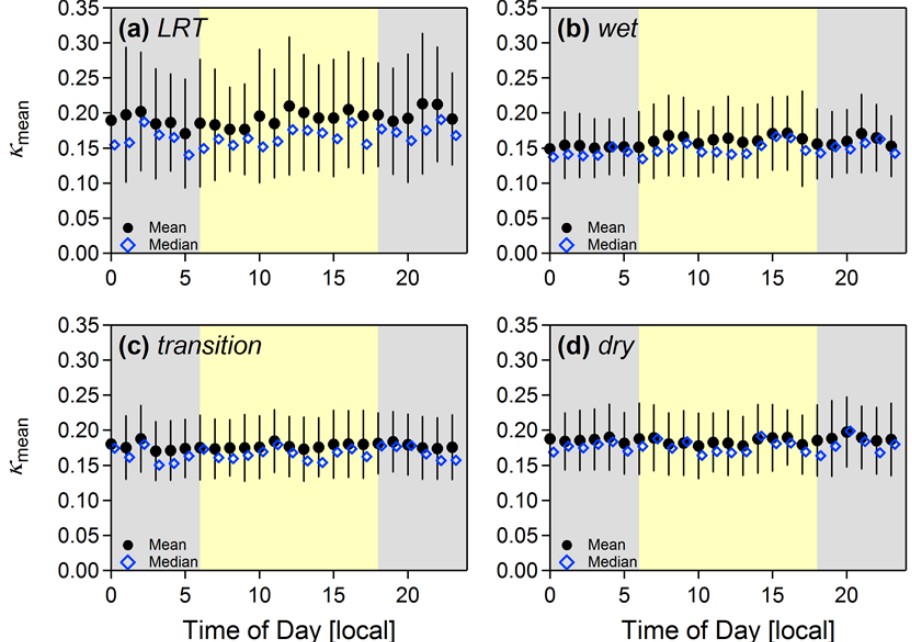

**Figure 7.** Diurnal cycles in hygroscopicity parameter $\kappa_{mean}$, subdivided into seasonal periods of interest as specified in Sect. 3.3. No diurnal trend is detectable throughout the year. Note that range of one standard deviation of $\kappa_{mean}$ around mean is surprisingly small given that long seasonal time periods and data from all $S$ levels have been averaged. Only perceptible difference is larger scattering during period with LRT influence (a). Grey and yellow shading indicates night and day.



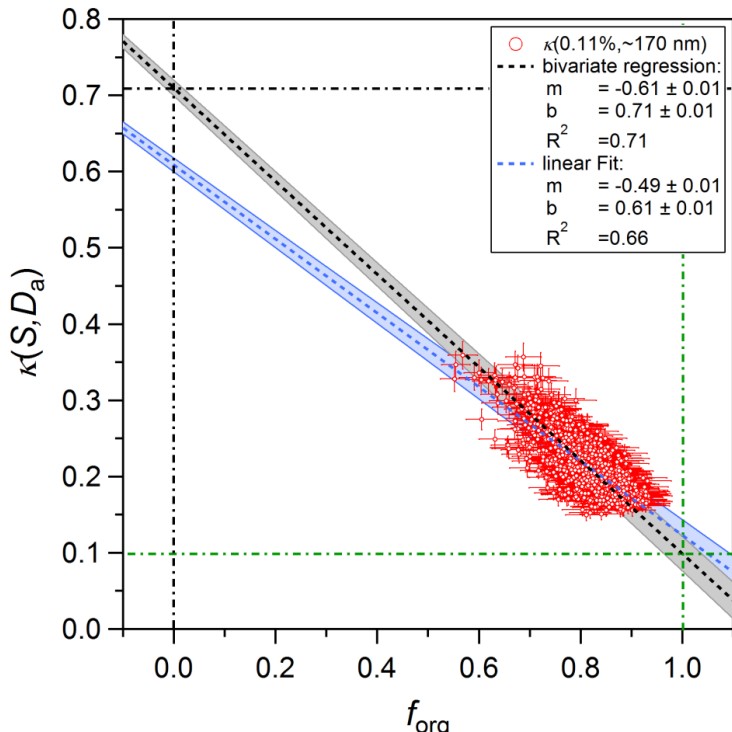

**Figure 8.** Correlation between $\kappa(0.11\%, \sim170\ nm)$ and the organic mass fraction $f_{org}$ determined by the ACSM during the dry season months. The data was fitted by a linear and a bivariate regression fit. Shading of the fit lines shows the standard error of the fit. The error bars of the data markers represent the experimental error, which is estimated as 5 % for $f_{org}$ and 10 % for $\kappa(0.11\%, \sim170\ nm)$.





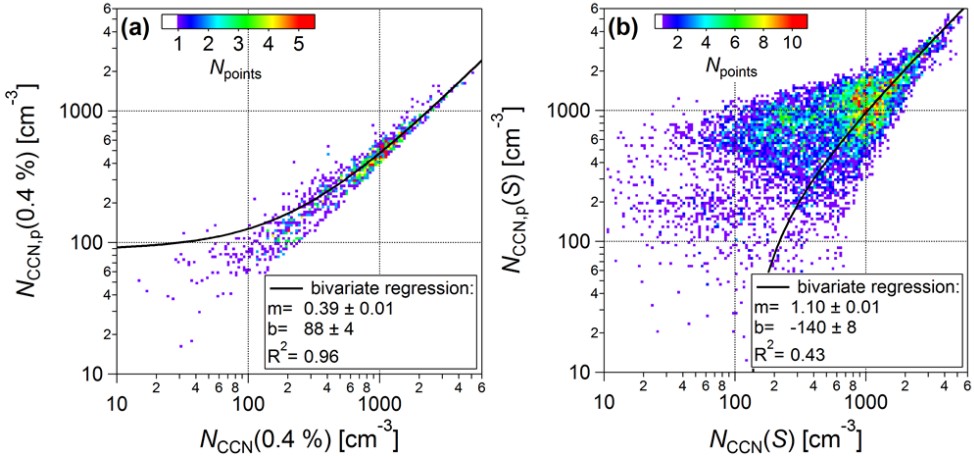

**Figure 9. Predicted *versus* measured CCN number concentrations calculated from (a) observed ratio $N_{CCN}(0.4\%)/N_{CN} = 0.36$ in Andreae et al. (2009) and (b) observed (biomass burning-related) excess CCN to excess CO ratios in M. L. Pöhlker et al. (2016b). The color code shows the number of data point falling into the pixel area, following Jurányi et al. (2011). The black line represents a bivariate regression fit of the data.**


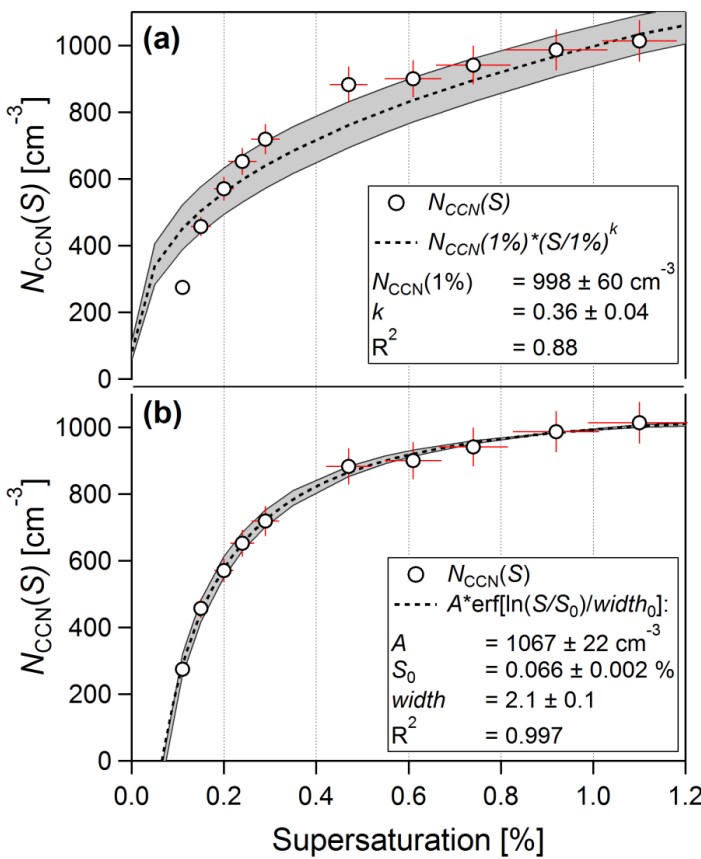

**Figure 10.** CCN spectrum (circular markers) averaged over entire measurement period and fitted with the classical Twomey power law fit (a) and an alternative error function fit (b). Error bars at markers represent the measurement error in $S$ and standard error in $N_{CCN}(S)$. Dashed line is fit function with grey shading as uncertainty of the fit.



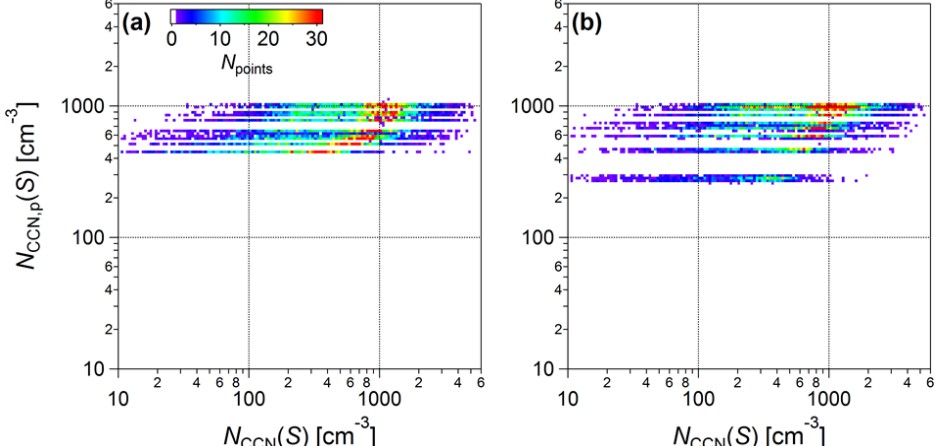

**Figure 11. Predicted *versus* measured CCN number concentrations based on the classical Twomey power law fit (a) and an alternative error function fit (b). Both predictions are based exclusively on the corresponding average fit functions functions (Fig. 10) without considering time resolved aerosol parameters. The color code shows the number of data point falling into the pixel area, following Jurányi et al. (2011). Predicted and measured CCN concentrations deviate significantly, showing the inherent limitations of the CCN spectra approach. Thus, no meaningful bivariate regression fit could be obtained here.**


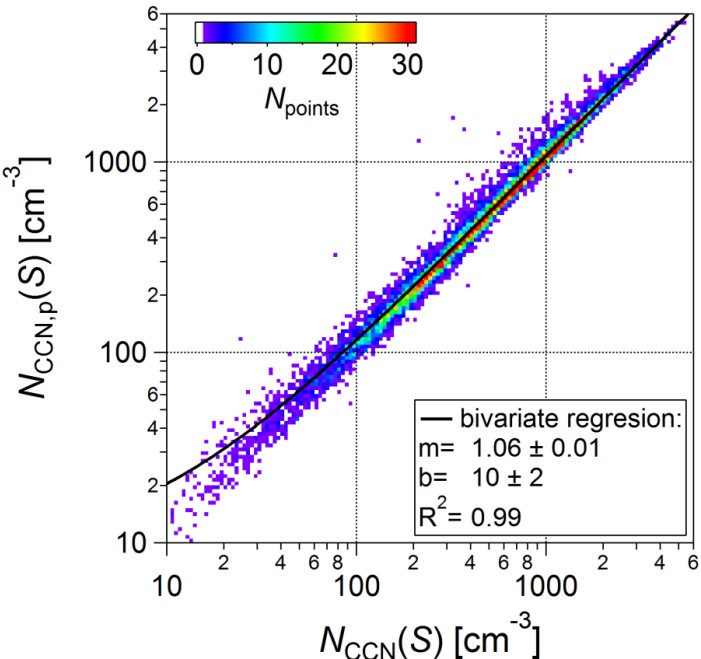

**Figure 12. Predicted *versus* measured CCN number concentrations, using the *κ-Köhler model* approach. This approach requires the following time-resolved aerosol input data: (i) time-resolved aerosol size spectra spanning the CCN-relevant range (e.g., SMPS) and (ii) annual average $\kappa$ values for the Aitken and accumulation size range ($\kappa_{Ait}$ = 0.14 and $\kappa_{Acc}$ = 0.22). The color code shows the number of data point falling into the pixel area, following Jurányi et al. (2011). The black line represents a bivariate regression fit of the data.**





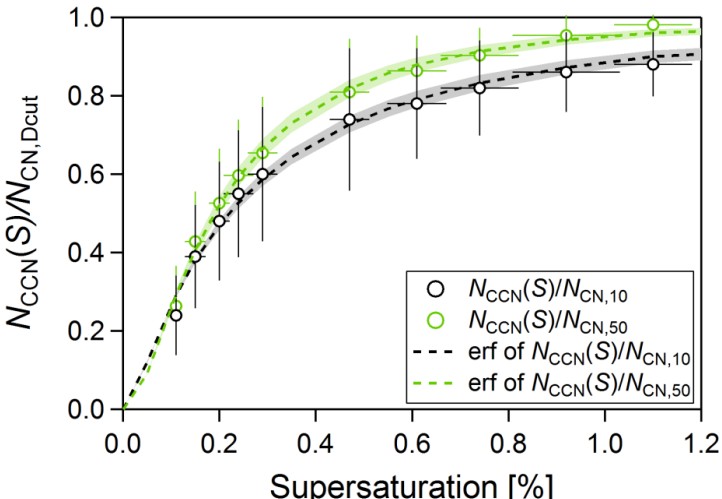

**Figure 13. CCN efficiency spectra averaged over entire measurement period for reference concentrations $N_{CN,10}$ and $N_{CN,50}$. Fit functions are error function fits (dashed line with shading is uncertainty of the fit). Error bars at markers represent the measurement error in $S$ and one standard deviation (not the standard error as in Fig. 10) in $N_{CCN}(S)/N_{CN,Dcut}$.**





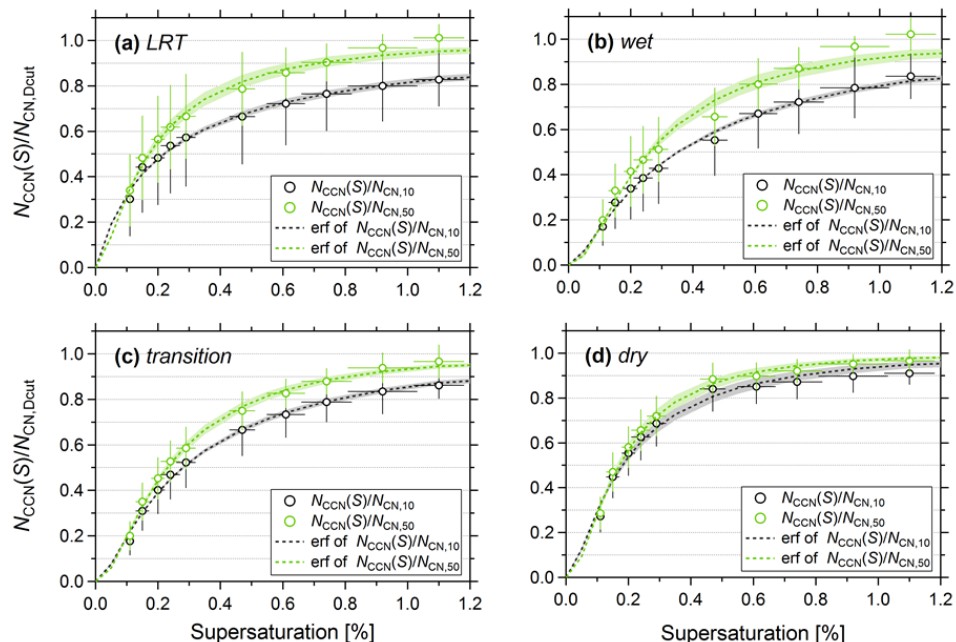

**Figure 14.** CCN efficiency spectra averaged over entire measurement period for reference concentrations $N_{CN,10}$ and $N_{CN,50}$ and subdivided into seasonal periods of interest as specified in Sect. 3.3. Fit functions are error function fits (dashed line with shading is uncertainty of the fit). Error bars at markers represent the measurement error in $S$ and one standard deviation (not the standard error as in Fig. 10) in $N_{CCN}(S)/N_{CN,Dcut}$.





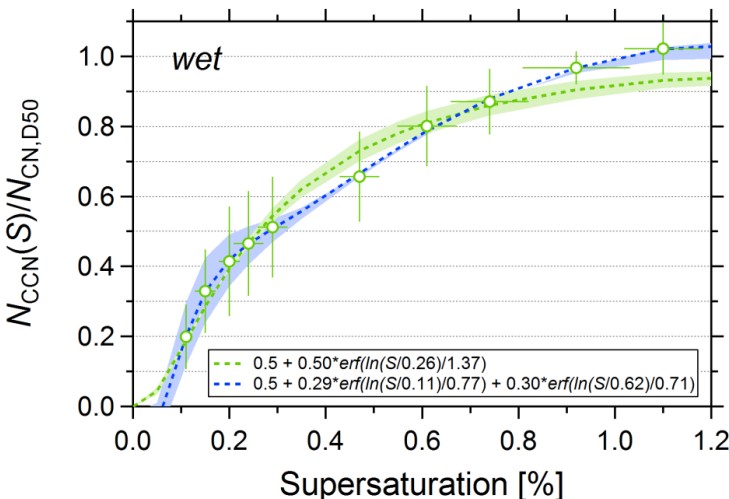

**Figure 15.** CCN efficiency spectrum for wet season scenario (Fig. 14b) with $N_{CN,50}$ as reference concentration. Experimental data has been fitted with single and double-erf fits (dashed lines with shading as uncertainty of the fits). Error bars at markers represent the measurement error in $S$ and one standard deviation in $N_{CCN}(S)/N_{CN,50}$.



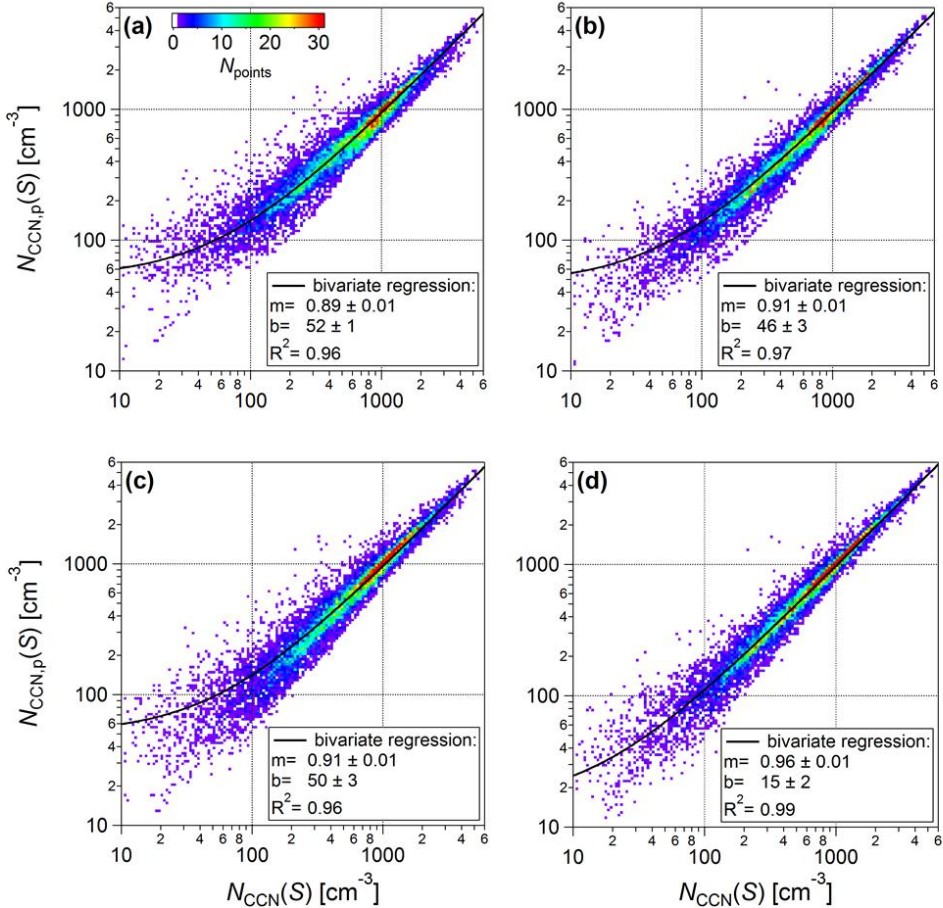

**Figure 16. Predicted *versus* measured CCN number concentrations, based on our novel parametrization using time-resolved aerosol number concentrations and annual average error function fits of CCN efficiency spectra. The panels show the following four variations of the parametrization: (a) erf fit of the annually averaged $N_{CCN}(S)/N_{CN,10}$ vs. $S$ efficiency plot, (b) erf fit of the annually averaged $N_{CCN}(S)/N_{CN,50}$ vs. $S$ efficiency plot, (c) erf fits of the $N_{CCN}(S)/N_{CN,10}$ vs. $S$ efficiency plot, resolved by seasons, and (d) erf fits of the $N_{CCN}(S)/N_{CN,50}$ vs. $S$ efficiency plot, resolved by seasons. This approach requires as input data: (i) a time series of total aerosol concentration (e.g., $N_{CN,10}$ from a CPC measurement or $N_{CN,50}$ as model output) and (ii) the parameters of the erf fit (e.g., as provided in Table 3). The color code shows the number of data point falling into the pixel area, following Jurányi et al. (2011). The black line represents a bivariate regression fit of the data.**