# Peer review of "Long-term observations of cloud condensation nuclei in the Amazon rain forest – Part 1: Aerosol size distribution, hygroscopicity, and new model parameterizations for CCN prediction"

_Atmospheric Chemistry and Physics, 2016_

## Referee Comment (RC1) · Anonymous Referee #1 · 1 Aug 2016

General comments: The manuscript discusses measurements of size-resolved atmospheric aerosol (particles size ranging from 20 to 245 nm) and cloud condensation nuclei (CCN) (supersaturation ranging from 0.11 to 1.10 %) concentrations as well as hygroscopicity at the remote Amazon Tall Tower Observatory (ATTO) in the central Amazon Basin over a full seasonal cycle from March 2014 to February 2015). This work finds little temporal variability (diurnal-seasonal) for hygroscopicity, but a pronounced temporal variability for CCN number concentrations, mostly driven by aerosol

particle number concentration and size distribution. A novel parametrization for CCN concentration is proposed to be applied in future modelling studies. The manuscript is generally well-written, the subject is important, especially for modelling studies, and I recommend it for publication in ACP.

Specific comments: In section 1.2 the authors discuss the hydrological regime of the Amazon forest. Onset and end of the rainy season in the central Amazon Basin, where the authors conducted their measurements, show the largest variations compared to other parts of the basin. Satellite-based outgoing longwave radiation (OLR) measurements and the low-level wind field show that in the central Amazon, onset is associated with anomalous anticyclones and enhanced trade winds in the Atlantic. In addition, there is an apparent association between sea surface temperature anomalies in the tropical Atlantic and Pacific and the onset and end of the rainy season in the central Amazon, in that a warm Pacific and cold Atlantic result in a delayed onset and early withdrawal. It is not clear in the text to what extent the clean (wet) and polluted (dry) seasons of the studied period (March 2014 to February 2015) are being impacted by large-scale atmospheric circulations, especially the temperatures of the Pacific and Atlantic and El Nino-Southern Oscillation.

In section 2.1 of the methodology the authors describe the characteristics of the measurement site and period. Although they justified in section 1.2 their preference for defining seasonality in terms of aerosol sources rather than meteorological variables, I suggest to include the monthly rainfall of the period of study overlayed the climatology of the central Amazon. Rainfall is a good indicator of how anomalous is the period of measurements. The monthly precipitation shown in the paper of Andreae et al. (2015), reported as a reference for an overview of the atmospheric conditions at the site, is incomplete for the year 2014.

Page 10 (lines 14-19): Using hygroscopicity parameter as reference, the authors state that particles' chemical composition is stable throughout the year and the maximum in CCN concentration during the dry season is mainly related to the overall increase in

aerosol concentration. In addition they considered their results as consistent with the previous result of Andreae et al. (2004) showing CCN efficiency (expressed as the ratio of CCN to CN) for the Amazonian wet and dry season aerosol. I do not particularly see condition to compare the two different studies. First, the finds of this study is unique in the sense it is the first time that we see a full year of CCN measurements in the Amazon. Therefore, there is no parallelism with the field campaigns CLAIRE-98 and SMOCC-2002, which are short campaigns. Second, they are measurement sites completely different. The only comparable CCN efficiencies are observed between SMOCC-2002 (cloud-processed smoke at altitude 2000 – 4500 m, cloud-processing might change the chemical composition and increase the hygroscopicity) and CLAIRE-98 (background ground-based measurements, naturally hygroscopic).

Page 10, lines 30-34: Even with a sparse occurrence of particles in the nucleation mode, did the authors find any seasonality in the number of nucleation episodes, such as the three representative days shown by Ortega et al. (2014) in Figure 9? How do the climate and forest of the central Amazon affect the absence of new particle formation, mainly when compared to other continental background locations such as the Manitou Experimental Forest Observatory (MEFO) described by Ortega et al. (2014)?

Page 11, lines 32-37 (discussion involving size dependence of hygroscopicity parameter): Why the values of hygroscopicity parameter, when averaged over the entire campaign (0.13±0.03 - Table 2 & Fig. 3) are practically constant in the Aitken mode? This is not observed in the accumulation mode. Could the differences be explained by the chemical composition or the cloud processing of the particles in the accumulation mode?

Page 13, lines 29-30: There are studies suggesting that aerosols from biomass burning is an ingredient to invigorate convective clouds. This is based on the fact that aerosols have a major impact on the microphysics of continental mixed-phase convective clouds. In addition to the solar heating suggested by the authors, could the aerosol effect also be a plausible explanation for the small Hoppel minimum and high cloud peak

supersaturation in the dry season?

Page 14, lines 28-32: The inclusion of the diurnal cycle of NCN, similar to Fig. 7, could enrich the discussion on the non detectable diurnal trend in the hygroscopicity parameter.

Page 34, Table 1: Hygroscopicity parameter is calculated as $0.13\pm0.03$ for both supersaturation 0.47% and 1.10%. However, looking at Fig. 01c, the parameter seem to be more dispersed at 1.10% than at 0.47% throughout the year. Is it correct a same std of 0.03 for both hygroscopic parameters?

Technical corrections: Page 10 (lines 8, 9 and 10) and wherever in the text: Please standardize the symbol for critical diameter Da(S) Page 17 and 19: It may be better to define a symbol for "width" in equations 7 and 8, considering its potential use in future articles.

Page 39: Fig. 1 is very useful to inform about seasonal trends in time series. However a plot of the diurnal cycle is missing for better understanding.

Page 41: Change "Aiken" to Aitken in the legend of Fig. 3.

Page 43: CCN activation curve at supersaturation of 0.47% shows strange values in the plots of the Fig. 5, including values of NCCN/NCN above 1.0.

---

## Referee Comment (RC2) · Anonymous Referee #2 · 2 Aug 2016

Long-term observations of atmospheric aerosol, cloud condensation nuclei concentration and hygroscopicity in the Amazon rain forest – Part 1: Size-resolved characterization and new model parameterizations for CCN prediction By Mira L. Pöhlker et al.

General comments This paper presents Size-resolved long-term measurements of atmospheric aerosol and cloud condensation nuclei (CCN) concentrations performed at the remote Amazon Tall Tower Observatory (ATTO). The study evaluates the evolution

of the aerosol hygroscopicity parameter during 10 months and 11 days, which can be said to cover a full seasonal cycle. The work presents the properties of aerosol hygroscopicities during four different period of interest, which represents contrasting aerosol conditions and sources, according to previous baseline works to the study. Furthermore, the authors evaluate the performance of parameterizations on the determination of CCN concentration considering the present database. New parameterization is also presented and compared to observations.

This work is interesting, well written and with an important subject. But I would recommend some small modifications before it goes for publication on ACP.

Specific comments:

I suggest that the period be referred as "full seasonal cycle" instead of "almost one year".

On page 11. section 3.2, line 21, we can read: "A close look reveals a gap between the activation curves for S = 0.47 % and S = 0.29 %, which corresponds to a jump in $\kappa(S,Da)$ (discussed below)." I could not see this gap! If we look closer the picture, we also can see that the intervals level used in supersaturation inside the CCNC jumps from ∼0.05% to ∼0.15%, which can explain the gap on the featured curves. So, what authors claim to correspond to a jump in hygroscopicities is, in fact, a result from the measurement. Is that right?

Page 14:

"Comparing the seasonal $\kappa(S,Da)$ size distributions in Fig. 6, it is obvious that the (seasonally averaged) $\kappa$Ait values in the Aitken mode size range are surprisingly stable between 0.13 and 0.14 throughout the whole year." This was already said at beginning of page 12 and also at page 10 (line 15). It was said three times in the text (and presented on table 1 too) that there is not an appreciable variation of hygroscopicity. Please, verify it. So, is Figure 7 really needed?

The parameterization of CCN spectra with constants (Twomey parameterization) has been used in many studies, most of them for short term observations. Though simple to carry out, it does not take into account any variation in the CCN loading, as was said in the text. It seems obvious to me, that the use of annual average for the constant used on the CCN spectra would result in overestimation of CCN concentration during the wet season, and in underestimation during the dry season. I would be more interesting if you could provide the constants for each season, instead of that for the whole year. Then the current section 3.5.3, as it is now, more weakens rather than strengthens the present work. Consider removing Figure 11.

Technical corrections: The text begins expressing supersaturation by "supersaturation S". Then it changes to "S", then to "S levels". Is it correct? Please check it.

---

## Author Comment (AC1) · 10 Oct 2016

We thank referee #2 for her/his positive review and recommendation that the study should be published in ACP after "some small modifications". The comments by referee #2 have been very constructive to improve several aspects of our study. The comments and our answers are listed below.

[2.1] Referee comment: I suggest that the period be referred as "full seasonal cycle"

[Figure]

instead of "almost one year".

Author Response: Agreed. We changed this on page 9 in line 35.

[2.2] Referee comment: On page 11. section 3.2, line 21, we can read: "A close look reveals a gap be-tween the activation curves for S = 0.47 % and S = 0.29 %, which corresponds to a jump in kappa(S,Da) (discussed below)." I could not see this gap! If we look closer the picture, we also can see that the inter-vals level used in supersaturation inside the CCNC jumps from ∼0.05% to ∼0.15%, which can explain the gap on the featured curves. So, what authors claim to correspond to a jump in hygroscopicities is, in fact, a result from the measurement. Is that right?

Author Response: We agree – our statement that the referee cited is indeed nonsense. We changed the corresponding section from:

"A close look reveals a gap between the activation curves for S = 0.47 % and S = 0.29 %, which corre-sponds to a jump in $\kappa$(S,Da) (discussed below). Moreover, the gap relates – in a way – to the bimodal size distribution and the characteristic Hoppel minimum (at 97 nm for the annual mean size distribution, see Table 2) between Aitken and accumulation mode, as S = 0.47 % represents the onset of significant activation in the Aitken mode size range."

to:

"The step from the activation curves at S = 0.47 % to S = 0.29 % relates to the position of the characteristic Hoppel minimum (at 97 nm for the annual mean size distribution, see Table 2) between Aitken and accumulation mode in the bimodal size distribution. Thus, the step to S = 0.47 % represents the onset of significant activation in the Aitken mode size range."

[2.3] Referee comment: Page 14: "Comparing the seasonal kappa(S,Da) size distribu-tions in Fig. 6, it is obvious that the (seasonally averaged) kappaAit values in the Aitken mode size range are surprisingly stable between 0.13 and 0.14 throughout the whole

year." This was already said at beginning of page 12 and also at page 10 (line 15). It was said three times in the text (and presented on table 1 too) that there is not an appreciable variation of hygroscopicity. Please, verify it. So, is Figure 7 really needed?

Author Response: The referee asks if "Figure 7 is really needed". Based on the context of the comment we assume that "Figure 7" is a typo and that the referee was referring to Fig. 6. Figure 6 is one of the key figures in this study and we think that it should not be omitted. We are convinced that it is justified to mention the small variation of $\kappa$Ait multiple times since this one of the key observations that supports our argumentation.

[2.4] Referee comment: The parameterization of CCN spectra with constants (Twomey parameteriza-tion) has been used in many studies, most of them for short term observations. Though simple to carry out, it does not take into account any variation in the CCN loading, as was said in the text. It seems obvious to me, that the use of annual average for the constant used on the CCN spectra would result in overestimation of CCN concentration during the wet season, and in underestimation during the dry season. I would be more interesting if you could provide the constants for each season, instead of that for the whole year. Then the current section 3.5.3, as it is now, more weakens rather than strengthens the present work. Consider removing Figure 11.

Author Response: The referee brings up a valid point. To implement his comment we made several modifications in the manuscript.

First, we conducted a seasonally resolved CCN prediction based on the Twomey and erf fit functions. The corresponding results have been added to Fig. 11.

Second, we added two further tables (as Table 6 and 7) into the manuscript, which summarize the Twomey and erf fit parameters for the annually average and seasonally resolved cases.

Third, we added the results from the seasonally resolved Twomey and erf fits to the overview Table 3.

Fourth, we modified the corresponding text section in Sect. 3.5.3 from:

"Figure 11a and b show the corresponding NCCN,p(S) versus NCCN(S) scatter plots. In general, parametrizations based on CCN spectra yield a mean state based on average concentrations (see fit parameters in Fig. 10) and ignore the temporal variability of the aerosol concentrations (Martins et al., 2009; Rose et al., 2010; Jurányi et al., 2011). On closer inspection, Table 3 shows that the erf fit allows somewhat better predictions (deviation of power law fit about 227 % versus 215 % for erf fit), which can be explained by the fact that the erf fit presents the experimental data more appropriately (compare Fig. 10). Overall, however, the power law fit and the erf fit approaches give rather poor correlations, due to the missing representation of the aerosol's temporal variability, which is an inherent limitation of the CCN spectra parametrization. It can be concluded that this parametrization requires a minimum of aerosol input data (i.e., only the parameters of the corresponding fit function), which explains its wide distribution in various modelling attempts. However, Fig. 10 and Table 3 show that this simplicity is clearly at the expense of the prediction accuracy."

to:

"Figure 11a and b show the corresponding NCCN,p(S) versus NCCN(S) scatter plots based on the annual mean CCN spectrum, using the Twomey and erf fits. In general, parametrizations based on CCN spectra yield a mean state based on average concentrations (see fit parameters in Fig. 10 as well as Table 5 and 6) and ignore the temporal variability of the aerosol's abundance (Martins et al., 2009; Rose et al., 2010; Jurányi et al., 2011). Table 3 shows that the erf fit allows somewhat better predictions (e.g., deviation of power law fit about 227 % versus 215 % for erf fit in case of annual mean and 80 % versus 75 % for the seasonally resolved case), which can be explained by the fact that the erf fit presents the experimental data more appropriately (compare with Fig. 10). Overall, however, the power law and erf fit approaches give rather poor correlations due to the missing representation of the aerosol's temporal variability. This is particularly obvious for the annual mean case since the total aerosol

abundance varies significantly between wet and dry season conditions. Accordingly, the CCN spectra parametrization, which operates with constants, predictably underestimates the dry season conditions and overestimates the wet season conditions. In addition to the analytical fit approaches for the annual mean spectrum (Fig. 11a and b) we conducted an analogous CCN prediction based on seasonally resolved CCN spectra (Fig. 11c and d). The prediction accuracy clearly improves (e.g., deviation of erf fit for annual mean case equals 215 % versus 75 % for seasonally resolved case; see Table 3). Figure 11 illustrates that the prediction accuracy of parametrizations, which rely on analytical fit functions of CCN spectra (i.e., Twomey, erf, and related functions), improves with decreasing variability of the aerosol population (e.g., for shorter periods with less variable aerosol properties). However, the missing representation of the aerosol's temporal variability remains to be an inherent limitation of the CCN spectra parametrization. It can be concluded that this parametrization requires a minimum of aerosol input data (i.e., only the parameters of the corresponding fit function), which explains its wide distribution in various modelling attempts. However, Fig. 11 and Table 3 show that this simplicity is clearly at the expense of the prediction accuracy."

[2.5] Referee comment: Technical corrections: The text begins expressing supersaturation by "super-saturation S". Then it changes to "S", then to "S levels". Is it correct? Please check it.

Author Response: This is correct. On page 7 in line 11, we introduced the symbol S for supersaturation. Throughout the text we then only refer to "S" or to "S levels", which is synonymously used for "supersaturations".

References:

Jurányi, Z., Gysel, M., Weingartner, E., Bukowiecki, N., Kammermann, L., and Baltensperger, U.: A 17 month climatology of the cloud condensation nuclei number concentration at the high alpine site Jungfraujoch, Journal of Geophysical Research: Atmospheres, 116, 10.1029/2010JD015199, 2011.

Martins, J. A., Dias, M., and Goncalves, F. L. T.: Impact of biomass burning aerosols on precipitation in the Amazon: A modeling case study, Journal of Geophysical Research-Atmospheres, 114, 19, 10.1029/2007jd009587, 2009.

Rose, D., Nowak, A., Achtert, P., Wiedensohler, A., Hu, M., Shao, M., Zhang, Y., Andreae, M. O., and Poschl, U.: Cloud condensation nuclei in polluted air and biomass burning smoke near the mega-city Guangzhou, China - Part 1: Size-resolved measurements and implications for the modeling of aerosol particle hygroscopicity and CCN activity, Atmospheric Chemistry and Physics, 10, 3365-3383, 2010.
* * *
[Figure]

Figure 11. Predicted versus measured CCN number concentrations based on the classical Twomey power law fit (a and c) and an alternative error function fit (b and d). The top row (a and b) represents the annually averaged cases, whereas the bottom row (c and d) represents parametrizations based on seasonally resolved CCN spectra. Both predictions are based exclusively on the corresponding average fit functions (i.e., the annually averaged CCN spectra in Fig. 10 and seasonally averaged CCN spectra, as specified in Table 6 and 7) without considering time resolved aerosol parameters. The color code shows the number of data point falling into the pixel area, following Jurányi et al. (2011). Predicted and measured CCN concentrations deviate significantly, showing the inherent limitations of the CCN spectra approach. For the annually averaged data (a and b) no meaningful bivariate regression fit could be obtained.

**Fig. 1.**

**Table 6. Twomey fit parameters describing CCN spectra $N_{\mathrm{CCN}}(S)$ *versus* $S$ as parametrization input data (compare Fig. 10 and 11a,c). Fit parameters are provided for annually averaged CCN spectra and resolved by seasons.**

| time period | $N_{\mathrm{CCN}}$ (1%) [cm$^{-3}$] | $K$ | $R^2$ |
|:---:|:---:|:---:|:---:|
| all | 998±60 | 0.36±0.04 | 0.88 |
| wet season | 289±7 | 0.57±0.03 | 0.98 |
| LRT period | 378±9 | 0.38±0.03 | 0.94 |
| transition | 970±40 | 0.49 ±0.05 | 0.94 |
| dry season | 1469±78 | 0.36 ±0.06 | 0.86 |

**Fig. 2.**

**Table 7.** Erf fit parameters describing CCN spectra $N_{CCN}(S)$ *versus* $S$ as parametrization input data (compare Fig. 10 and 11b,d). Fit parameters are provided for annually averaged CCN spectra and resolved by seasons.

| time period | $A$ [cm$^{-3}$] | $S_0$ [%] | $w_0$ | $R^2$ |
|---|---|---|---|---|
| all | 1067±22 | 0.07±0.01 | 2.1±0.1 | 0.99 |
| wet season | 340±30 | 0.08±0.01 | 2.9 ±0.2 | 0.97 |
| LRT period | 532±72 | 0.04±0.01 | 4.5±1.0 | 0.98 |
| transition | 1180±37 | 0.07±0.01 | 3.0 ±0.2 | 0.99 |
| dry season | 1430±24 | 0.07±0.01 | 1.8 ±0.1 | 0.99 |

**Fig. 3.**

[Figure]

---

## Author Comment (AC2) · 10 Oct 2016

We thank Referee #1 for the pertinent comments and suggestions that have helped us to improve the quality of our manuscript. The referees' comments and our responses are outlined in detail below:

[1.1] Referee comment: In section 1.2 the authors discuss the hydrological regime of the Amazon forest. Onset and end of the rainy season in the central Amazon Basin,

where the authors conducted their measurements, show the largest variations compared to other parts of the basin. Satellite-based outgoing longwave radiation (OLR) measurements and the low-level wind field show that in the central Amazon, onset is associated with anomalous anticyclones and enhanced trade winds in the Atlantic. In addition, there is an apparent association between sea surface temperature anomalies in the tropical Atlantic and Pacific and the onset and end of the rainy season in the central Amazon, in that a warm Pacific and cold Atlantic result in a delayed onset and early withdrawal. It is not clear in the text to what extent the clean (wet) and polluted (dry) seasons of the studied period (March 2014 to February 2015) are being impacted by large-scale atmospheric circulations, especially the temperatures of the Pacific and Atlantic and El Nino-Southern Oscillation.

Author Response: We agree that discussing (potential) anomalies in the hydrological regime for the studied period in 2014/15 due to teleconnections with the ocean surface temperatures would strengthen the paper. Therefore, we modified and adjusted the following parts in the paper:

First, we changed the footnote number 1 on page 3 from:

"Note that this definition of the seasons in the central Amazon is oriented on the seasonality in aerosol sources and prevalence rather than the meteorological conditions. For example, the 'meteorological wet season' typically has its core period in February (maximum in precipitation), whereas the 'pollution-defined wet season' has its core period in April/May (e.g., minimum in CO and BC concentrations) (Andreae et al, 2015)."

to:

"The Amazonian seasons are mostly defined meteorologically with respect to precipitation data (Fu et al., 2001; Fernandes et al., 2015). Note that we use in this study a slightly different definition of the seasons in the central Amazon based on meteorological and aerosol data to emphasize the seasonality in aerosol sources and prevalence. For example, the 'meteorological wet season' typically has its core period in February

(maximum in precipitation), whereas the 'pollution-defined wet season' has its core period in April/May (e.g., minimum in CO and BC concentrations) (Andreae et al, 2015)."

Second, we added the following text section on page 10 in line 5:

"Figure 1a presents precipitation data from satellite and in situ measurements at the ATTO site to illustrate the meteorological seasonality for the measurement period. The precipitation rates in the Amazon Basin can show pronounced anomalies due to teleconnections with the Atlantic and/or Pacific sea surface temperatures (SST) (Fu et al., 2001; Fernandes et al., 2015). The most prominent example here is the El Niño-Southern Oscillation (ENSO) and its various impacts on the Amazonian ecosystem (e.g., Asner et al., 2000; Ronchail et al., 2002). For the measurement period, the Oceanic Niño Index (ONI) ranged between -0.4 and 0.6 °C, confirming that only towards the end of the measurement period a slightly positive anomaly was observed. In Fig. 1a, satellite data from the tropical rainfall measurement mission (TRMM) are presented for the area around the ATTO site. The TRMM data is provided for an extended time period (Jan 1998 until June 2016) and, in comparison, for the CCN measurement period (Mar 2014 until Feb 2015). This comparison shows that the 2014/15 precipitation rates do not deviate substantially from the 18-year average and, thus, further confirms that the measurement period can be regarded as a 'typical' year with 'typical' seasons and no pronounced hydrological anomalies."

Third, precipitation data has been added to Fig. 1 to illustrate the absence of potential hydrological anomalies in the measurement period.

[1.2] Referee comment: In section 2.1 of the methodology, the authors describe the characteristics of the measurement site and period. Although they justified in section 1.2 their preference for defining seasonality in terms of aerosol sources rather than meteorological variables, I suggest to include the monthly rainfall of the period of study overlayed the climatology of the central Amazon. Rainfall is a good indicator of how anomalous is the period of measurements. The monthly precipitation shown in the paper of Andreae et al. (2015), reported as a reference for an overview of the atmospheric conditions at the site, is incomplete for the year 2014.

Author Response: We agree that precipitation data will help to clarify the meteorological seasonality. We added to Fig. 1 precipitation data from the TRMM satellite mission and from in situ measurements at the ATTO site – see our response to comment [1.1]. Moreover, we would like to point out that the seasonality in precipitation will be discussed in more detail in part 2 of this study and, therefore, complete the picture.

[1.3] Referee comment: Page 10 (lines 14-19): Using hygroscopicity parameter as reference, the au-thors state that particles' chemical composition is stable throughout the year and the maximum in CCN concentration during the dry season is mainly related to the overall increase in aerosol concentration. In addition they considered their results as consistent with the previous result of Andreae et al. (2004) showing CCN efficiency (expressed as the ratio of CCN to CN) for the Amazonian wet and dry season aerosol. I do not particularly see condition to compare the two different studies. First, the finds of this study is unique in the sense it is the first time that we see a full year of CCN measurements in the Amazon. Therefore, there is no parallelism with the field campaigns CLAIRE-98 and SMOCC-2002, which are short campaigns. Second, they are measurement sites completely different. The only comparable CCN efficiencies are observed between SMOCC-2002 (cloud-processed smoke at altitude 2000 – 4500 m, cloud-processing might change the chemical composition and increase the hygroscopicity) and CLAIRE-98 (background ground-based measurements, naturally hygroscopic).

Author Response: Agreed. We removed the following sentence from the text:

"Furthermore, this observation is consistent with the previously reported similarity between the CCN efficiency of Amazonian wet and dry season aerosol (Andreae et al., 2004)."

[1.4] Referee comment: Page 10, lines 30-34: Even with a sparse occurrence of particles in the nuclea-tion mode, did the authors find any seasonality in the number of nu-cleation episodes, such as the three representative days shown by Ortega et al. (2014) in Figure 9? How do the climate and forest of the central Amazon affect the absence of new particle formation, mainly when compared to other continental background locations such as the Manitou Experimental Forest Observatory (MEFO) described by Ortega et al. (2014)?

Author Response: The referee points at an interesting aspect, which is the sparse occurrence of nucleation mode particles in the Amazon. The question whether there is any seasonality in the frequency of the nucleation mode events is not trivial to answer. It requires a reliable discrimination of event versus non-event cases and, furthermore, a systematic statistical approach to extract seasonal trends. A detailed analysis on the abundance, properties, and seasonality of the rare nucleation mode events at the ATTO site is subject of a study that is currently prepared for publication. We added the following statement into the text (page 11, line 6):

"A systematic study on the abundance, properties, and seasonality of the sparse nucleation mode bursts in the central Amazon is subject of an upcoming study."

[1.5] Referee comment: Page 11, lines 32-37 (discussion involving size dependence of hygroscopicity parameter): Why the values of hygroscopicity parameter, when aver-aged over the entire campaign ($0.13\pm0.03$ - Table 2 & Fig. 3) are practically constant in the Aitken mode? This is not observed in the accumulation mode. Could the differences be explained by the chemical composition or the cloud pro-cessing of the particles in the accumulation mode?

Author Response: The referee points at an interesting aspect. The origin and nature of Aitken mode particles in the Amazon Basin still raises a number of open questions. In a recent study, Wang et al. (2016) propose that Aitken mode particles originate from nucleation in the free troposphere and are frequently injected into the boundary layer by down-drafted air masses in connection with strong rain. Pöschl et al. (2010)

showed in the context of the AMAZE-08 campaign that the Aitken mode size range almost exclusively consists of organic constituents, whereas the accumulation mode contains a certain amount of sulfates (and probably also other inorganic ingredients) beside its dominant organic fraction. However, there is so far only sparse information on the chemical composition of Aitken mode particles available. In this sense the data in our study is unique since it confirms for a long time period that the Aitken mode aerosol population consists of almost entirely organic constituents (indirectly via the hygroscopicity properties). Moreover, we find that the accumulation mode showing elevated hygroscopicity is in agreement with observations by Pöschl et al. (2010). If and how cloud processing influences the distinct differences in chemical composition of Aitken and accumulation modes is beyond the scope of this paper and will be subject of future studies. Some further information on the abundance and hygroscopicity of Aitken and accumulation mode particles for specific events will be addressed in the part 2 paper of this study (M. L. Pöhlker et al., 2016).

[1.6] Referee comment: Page 13, lines 29-30: There are studies suggesting that aerosols from biomass burning is an ingredient to invigorate convective clouds. This is based on the fact that aerosols have a major impact on the microphysics of continental mixed-phase convective clouds. In addition to the solar heating suggested by the authors, could the aerosol effect also be a plausible explanation for the small Hoppel minimum and high cloud peak supersaturation in the dry season?

Author Response: We agree and added the corresponding statement on page 13 line 30 from:

"A plausible explanation for the comparably small DH and high Scloud(DH,$\kappa$) in the dry season could be the invigorated updraft regimes due to stronger solar heating."

to:

"A plausible explanation for the comparatively small DH and high Scloud(DH,$\kappa$) in the dry season could be invigorated updraft regimes in the convective clouds. This invigoration could be caused by the stronger solar heating during the dry season and/or the increased aerosol load under bio-mass burning impacted conditions, as suggested previously (Andreae et al., 2004; Rosenfeld et al., 2008)."

[1.7] Referee comment: Page 14, lines 28-32: The inclusion of the diurnal cycle of NCN, similar to Fig. 7, could enrich the discussion on the non detectable diurnal trend in the hygroscopicity parameter.

Author Response: We agree. The diurnal cycle of the total aerosol number concentration has been included in Fig. 7.

Furthermore, we added the following statement on page 14 in line 32:

"For comparison, the diurnal cycles in NCN concentration have been added to Fig. 7, which con-firm the absence of strong diurnal variations in the aerosol population."

[1.8] Referee comment: Page 34, Table 1: Hygroscopicity parameter is calculated as $0.13\pm0.03$ for both supersaturation 0.47% and 1.10%. However, looking at Fig. 01c, the parameter seem to be more dispersed at 1.10% than at 0.47% throughout the year. Is it correct a same std of 0.03 for both hygroscopic parameters?

Author Response: The same standard deviation of 0.03 for both cases is correct. Please note that the error bars in Fig. 1c represent the experimental error in $\kappa(S,Da)$, derived from the experimental error in S. The $\kappa(S,Da)$ values for every S reported in Table 1 are given as mean $\pm$ one standard deviation.

[1.9] Referee comment: Technical corrections: Page 10 (lines 8, 9 and 10) and wherever in the text: Please standardize the symbol for critical diameter Da(S) Page 17 and 19: It may be better to define a symbol for "width" in equations 7 and 8, considering its potential use in future articles.

Author Response: Done. We standardized all symbols of Da(S) (replacing the Da). Moreover, we followed the referees suggestion and replace "width0" in the context of CCN spectra and "width1" in the context of CCN efficiency spectra by "w0" and "w1"

throughout the text.

[1.10] Referee comment: Page 39: Fig. 1 is very useful to inform about seasonal trends in time series. However a plot of the diurnal cycle is missing for better understanding.

Author Response: In the new version of Fig. 8 the diurnal cycles in $\kappa$mean and NCN, resolved by seasons, are shown. More specific diurnal cycles of aerosol concentration in defined size ranges will be subject of upcoming studies.

[1.11] Referee comment: Page 41: Change "Aiken" to Aitken in the legend of Fig. 3.

Author Response: Done. This has been changed:

[1.12] Referee comment: Page 43: CCN activation curve at supersaturation of 0.47% shows strange values in the plots of the Fig. 5, including values of NCCN/NCN above 1.0.

Author Response: Agreed. To clarify this aspect, we added the following statement to Sect. 2.3:

"Throughout this study, we observed a slight systematic deviation of the results for the supersaturation S = 0.47 %. This effect can be seen for example in MAF(0.47%) values exceeding unity in Fig. 1 and NCCN(0.47%,D)/NCN(D) values exceeding unity in Fig. 5. The effect persists even after applying all aforementioned corrections to the data and is most pronounced during the dry season. Yet, we did not find any evidence of this data being erroneous, we decided to keep it in the study."

References:

[revised manuscript text omitted]

---

## Author Response (AR1)

Response to the referees (M. L. Pöhlker et al., Long-term observations of cloud condensation nuclei in the Amazon rain forest – Part 1: Aerosol size distribution, hygroscopicity, and new model parametrization for CCN prediction, ACP-2016-519)

We thank Referee #1 for the pertinent comments and suggestions that have helped us to improve the quality of our manuscript. The referees' comments and our responses are outlined in detail below:

[1.1]    Referee comment: In section 1.2 the authors discuss the hydrological regime of the Amazon forest. Onset and end of the rainy season in the central Amazon Basin, where the authors conducted their measurements, show the largest variations compared to other parts of the basin. Satellite-based outgoing longwave radiation (OLR) measurements and the low-level wind field show that in the central Amazon, onset is associated with anomalous anticyclones and enhanced trade winds in the Atlantic. In addition, there is an apparent association between sea surface temperature anomalies in the tropical Atlantic and Pacific and the onset and end of the rainy season in the central Amazon, in that a warm Pacific and cold Atlantic result in a delayed onset and early withdrawal. It is not clear in the text to what extent the clean (wet) and polluted (dry) seasons of the studied period (March 2014 to February 2015) are being impacted by large-scale atmospheric circulations, especially the temperatures of the Pacific and Atlantic and El Nino-Southern Oscillation.

Author Response: We agree that discussing (potential) anomalies in the hydrological regime for the studied period in 2014/15 due to teleconnections with the ocean surface temperatures would strengthen the paper. Therefore, we modified and adjusted the following parts in the paper:

First, we changed the footnote number 1 on page 3 from:

[revised manuscript text omitted]

[1.2]    Referee comment: In section 2.1 of the methodology, the authors describe the characteristics of the measurement site and period. Although they justified in section 1.2 their preference for defining seasonality in terms of aerosol sources rather than meteorological variables, I suggest to include the monthly rainfall of the period of study overlayed the climatology of the central Amazon. Rainfall is a good indicator of how anomalous is the period of measurements. The monthly precipitation shown in the paper of Andreae et al. (2015), reported as a reference for an overview of the atmospheric conditions at the site, is incomplete for the year 2014.

Author Response: We agree that precipitation data will help to clarify the meteorological seasonality. We added to Fig. 1 precipitation data from the TRMM satellite mission and from *in situ* measurements at the ATTO site – see our response to comment [1.1]. Moreover, we would like to point out that the seasonality in precipitation will be discussed in more detail in part 2 of this study and, therefore, complete the picture.

[1.3]    Referee comment: Page 10 (lines 14-19): Using hygroscopicity parameter as reference, the authors state that particles' chemical composition is stable throughout the year and the maximum in CCN concentration during the dry season is mainly related to the overall increase in aerosol concentration. In addition they considered their results as consistent with the previous result of Andreae et al. (2004) showing CCN efficiency (expressed as the ratio of CCN to CN) for the Amazonian wet and dry season aerosol. I do not particularly see condition to compare the two different studies. First, the finds of this study is unique in the sense it is the first time that we see a full year of CCN measurements in the Amazon. Therefore, there is no parallelism with the field campaigns CLAIRE-98 and SMOCC-2002, which are short campaigns. Second, they are measurement sites completely different. The only comparable CCN efficiencies are observed between SMOCC-2002 (cloud-processed smoke at altitude 2000 – 4500 m, cloud-processing might change the chemical composition and increase the hygroscopicity) and CLAIRE-98 (background ground-based measurements, naturally hygroscopic).

Author Response: Agreed. We removed the following sentence from the text:

"Furthermore, this observation is consistent with the previously reported similarity between the CCN efficiency of Amazonian wet and dry season aerosol (Andreae et al., 2004)."

[1.4]    Referee comment: Page 10, lines 30-34: Even with a sparse occurrence of particles in the nucleation mode, did the authors find any seasonality in the number of nucleation episodes, such as the three representative days shown by Ortega et al. (2014) in Figure 9? How do the climate and forest of the central Amazon affect the absence of new particle formation, mainly when compared to other continental background locations such as the Manitou Experimental Forest Observatory (MEFO) described by Ortega et al. (2014)?

Author Response: The referee points at an interesting aspect, which is the sparse occurrence of nucleation mode particles in the Amazon. The question whether there is any seasonality in the frequency of the nucleation mode events is not trivial to answer. It requires a reliable discrimination of event *versus* non-event cases and, furthermore, a systematic statistical approach to extract seasonal trends. A detailed analysis on the abundance, properties, and seasonality of the rare nucleation mode events at the ATTO site is subject of a study that is currently prepared for publication. We added the following statement into the text (page 11, line 6):

"A systematic study on the abundance, properties, and seasonality of the sparse nucleation mode bursts in the central Amazon is subject of an upcoming study."

[1.5]   Referee comment: Page 11, lines 32-37 (discussion involving size dependence of hygroscopicity parameter): Why the values of hygroscopicity parameter, when averaged over the entire campaign (0.13±0.03 - Table 2 & Fig. 3) are practically constant in the Aitken mode? This is not observed in the accumulation mode. Could the differences be explained by the chemical composition or the cloud processing of the particles in the accumulation mode?

Author Response: The referee points at an interesting aspect. The origin and nature of Aitken mode particles in the Amazon Basin still raises a number of open questions. In a recent study, Wang et al. (2016) propose that Aitken mode particles originate from nucleation in the free troposphere and are frequently injected into the boundary layer by down-drafted air masses in connection with strong rain. Pöschl et al. (2010) showed in the context of the AMAZE-08 campaign that the Aitken mode size range almost exclusively consists of organic constituents, whereas the accumulation mode contains a certain amount of sulfates (and probably also other inorganic ingredients) beside its dominant organic fraction. However, there is so far only sparse information on the chemical composition of Aitken mode particles available. In this sense the data in our study is unique since it confirms for a long time period that the Aitken mode aerosol population consists of almost entirely organic constituents (indirectly via the hygroscopicity properties). Moreover, we find that the accumulation mode showing elevated hygroscopicity is in agreement with observations by Pöschl et al. (2010). If and how cloud processing influences the distinct differences in chemical composition of Aitken and accumulation modes is beyond the scope of this paper and will be subject of future studies. Some further information on the abundance and hygroscopicity of Aitken and accumulation mode particles for specific events will be addressed in the part 2 paper of this study (M. L. Pöhlker et al., 2016).

[1.6]   Referee comment: Page 13, lines 29-30: There are studies suggesting that aerosols from biomass burning is an ingredient to invigorate convective clouds. This is based on the fact that aerosols have a major impact on the microphysics of continental mixed-phase convective clouds. In addition to the solar heating suggested by the authors, could the aerosol effect also be a plausible explanation for the small Hoppel minimum and high cloud peak supersaturation in the dry season?

Author Response: We agree and added the corresponding statement on page 13 line 30 from:

> "A plausible explanation for the comparably small $D_H$ and high $S_{cloud}(D_H,\kappa)$ in the dry season could be the invigorated updraft regimes due to stronger solar heating."

to:

> "A plausible explanation for the comparatively small $D_H$ and high $S_{cloud}(D_H,\kappa)$ in the dry season could be invigorated updraft regimes in the convective clouds. This invigoration could be caused by the stronger solar heating during the dry season and/or the increased aerosol load under biomass burning impacted conditions, as suggested previously (Andreae et al., 2004; Rosenfeld et al., 2008)."

[1.7]   Referee comment: Page 14, lines 28-32: The inclusion of the diurnal cycle of NCN, similar to Fig. 7, could enrich the discussion on the non detectable diurnal trend in the hygroscopicity parameter.

Author Response: We agree. The diurnal cycle of the total aerosol number concentration has been included in Fig. 7:

[Figure]

**Figure 7. Diurnal cycles in hygroscopicity parameter $\kappa_{\text{mean}}$ and total aerosol number concentration $N_{\text{CN}}$ subdivided into seasonal periods of interest as specified in Sect. 3.3. No diurnal trend is detectable throughout the year. Note that range of one standard deviation of $\kappa_{\text{mean}}$ around mean is surprisingly small given that long seasonal time periods and data from all $S$ levels have been averaged. Only perceptible difference is larger scattering during period with LRT influence (a). Grey and yellow shading indicates night and day.**

Furthermore, we added the following statement on page 14 in line 32:

> "For comparison, the diurnal cycles in $N_{\text{CN}}$ concentration have been added to Fig. 7, which confirm the absence of strong diurnal variations in the aerosol population."

[1.8]    Referee comment: Page 34, Table 1: Hygroscopicity parameter is calculated as 0.13±0.03 for both supersaturation 0.47% and 1.10%. However, looking at Fig. 01c, the parameter seem to be more dispersed at 1.10% than at 0.47% throughout the year. Is it correct a same std of 0.03 for both hygroscopic parameters?

Author Response: The same standard deviation of 0.03 for both cases is correct. Please note that the error bars in Fig. 1c represent the *experimental error* in $\kappa(S,D_a)$, derived from the experimental error in $S$. The $\kappa(S,D_a)$ values for every $S$ reported in Table 1 are given as mean $\pm$ *one standard deviation*.

[1.9]    Referee comment: Technical corrections: Page 10 (lines 8, 9 and 10) and wherever in the text: Please standardize the symbol for critical diameter Da(S) Page 17 and 19: It may be better to define a symbol for "width" in equations 7 and 8, considering its potential use in future articles.

Author Response: Done. We standardized all symbols of $D_a(S)$ (replacing the $D_a$). Moreover, we followed the referees suggestion and replace „*width*$_0$" in the context of CCN spectra and „*width*$_1$" in the context of CCN efficiency spectra by "$w_0$" and "$w_1$" throughout the text.

[1.10]   Referee comment: Page 39: Fig. 1 is very useful to inform about seasonal trends in time series. However a plot of the diurnal cycle is missing for better understanding.

Author Response: In the new version of Fig. 8 the diurnal cycles in $\kappa_{\text{mean}}$ and $N_{\text{CN}}$, resolved by seasons, are shown. More specific diurnal cycles of aerosol concentration in defined size ranges will be subject of upcoming studies.

[1.11]   Referee comment: Page 41: Change "Aiken" to Aitken in the legend of Fig. 3.

Author Response: Done. This has been changed:

[1.12]   Referee comment: Page 43: CCN activation curve at supersaturation of 0.47% shows strange values in the plots of the Fig. 5, including values of NCCN/NCN above 1.0.

Author Response: Agreed. To clarify this aspect, we added the following statement to Sect. 2.3:

> "Throughout this study, we observed a slight systematic deviation of the results for the supersaturation S = 0.47 %. This effect can be seen for example in $MAF(0.47\%)$ values exceeding unity in Fig. 1 and $N_{\text{CCN}}(0.47\%, D)/N_{\text{CN}}(D)$ values exceeding unity in Fig. 5. The effect persists even after applying all aforementioned corrections to the data and is most pronounced during the dry season. Yet, we did not find any evidence of this data being erroneous, we decided to keep it in the study."

Author Response: We agree – our statement that the referee cited is indeed nonsense. We changed the corresponding section from:

"A close look reveals a gap between the activation curves for $S = 0.47$ % and $S = 0.29$ %, which corresponds to a jump in $\kappa(S,D_a)$ (discussed below). Moreover, the gap relates – in a way – to the bimodal size distribution and the characteristic Hoppel minimum (at 97 nm for the annual mean size distribution, see Table 2) between Aitken and accumulation mode, as $S = 0.47$ % represents the onset of significant activation in the Aitken mode size range."

to:

"The step from the activation curves at $S = 0.47$ % to $S = 0.29$ % relates to the position of the characteristic Hoppel minimum (at 97 nm for the annual mean size distribution, see Table 2) between Aitken and accumulation mode in the bimodal size distribution. Thus, the step to $S = 0.47$ % represents the onset of significant activation in the Aitken mode size range."

[2.3]    Referee comment: Page 14: "Comparing the seasonal kappa(S,Da) size distributions in Fig. 6, it is obvious that the (seasonally averaged) kappaAit values in the Aitken mode size range are surprisingly stable between 0.13 and 0.14 throughout the whole year." This was already said at beginning of page 12 and also at page 10 (line 15). It was said three times in the text (and presented on table 1 too) that there is not an appreciable variation of hygroscopicity. Please, verify it. So, is Figure 7 really needed?

Author Response: The referee asks if "Figure 7 is really needed". Based on the context of the comment we assume that "Figure 7" is a typo and that the referee was referring to Fig. 6. Figure 6 is one of the key figures in this study and we think that it should not be omitted. We are convinced that it is justified to mention the small variation of $\kappa_{Ait}$ multiple times since this one of the key observations that supports our argumentation.

[2.4]    Referee comment: The parameterization of CCN spectra with constants (Twomey parameterization) has been used in many studies, most of them for short term observations. Though simple to carry out, it does not take into account any variation in the CCN loading, as was said in the text. It seems obvious to me, that the use of annual average for the constant used on the CCN spectra would result in overestimation of CCN concentration during the wet season, and in underestimation during the dry season. I would be more interesting if you could provide the constants for each season, instead of that for the whole year. Then the current section 3.5.3, as it is now, more weakens rather than strengthens the present work. Consider removing Figure 11.

Author Response: The referee brings up a valid point. To implement his comment we made several modifications in the manuscript.

First, we conducted a *seasonally resolved* CCN prediction based on the Twomey and erf fit functions. The corresponding results have been added to Fig. 11:

[Figure]

**Figure 11. Predicted *versus* measured CCN number concentrations based on the classical Twomey power law fit (a and c) and an alternative error function fit (b and d). The top row (a and b) represents the annually averaged cases, whereas the bottom row (c and d) represents parametrizations based on seasonally resolved CCN spectra. Both predictions are based exclusively on the corresponding average fit functions (i.e., the annually averaged CCN spectra in Fig. 10 and seasonally averaged CCN spectra, as specified in Table 6 and 7) without considering time resolved aerosol parameters. The color code shows the number of data point falling into the pixel area, following Jurányi et al. (2011). Predicted and measured CCN concentrations deviate significantly, showing the inherent limitations of the CCN spectra approach. For the annually averaged data (a and b) no meaningful bivariate regression fit could be obtained.**

Second, we added two further tables (as Table 6 and 7) into the manuscript, which summarize the Twomey and erf fit parameters for the annually average and seasonally resolved cases:

**Table 6. Twomey fit parameters describing CCN spectra $N_{CCN}(S)$ *versus* S as parametrization input data (compare Fig. 10 and 11a,c). Fit parameters are provided for annually averaged CCN spectra and resolved by seasons.**

| time period | $N_{CCN}$ (1%) [cm$^{-3}$] | $k$ | $R^2$ |
|---|---|---|---|
| all | 998±60 | 0.36±0.04 | 0.88 |
| wet season | 289±7 | 0.57±0.03 | 0.98 |
| LRT period | 378±9 | 0.38±0.03 | 0.94 |
| transition | 970±40 | 0.49 ±0.05 | 0.94 |
| dry season | 1469±78 | 0.36 ±0.06 | 0.86 |

**Table 7. Erf fit parameters describing CCN spectra $N_{CCN}(S)$ *versus* S as parametrization input data (compare Fig. 10 and 11b,d). Fit parameters are provided for annually averaged CCN spectra and resolved by seasons.**

| time period | $A$ [cm$^{-3}$] | $S_0$ [%] | $w_0$ | $R^2$ |
|---|---|---|---|---|
| all | 1067±22 | 0.07±0.01 | 2.1±0.1 | 0.99 |
| wet season | 340±30 | 0.08±0.01 | 2.9 ±0.2 | 0.97 |
| LRT period | 532±72 | 0.04±0.01 | 4.5±1.0 | 0.98 |
| transition | 1180±37 | 0.07±0.01 | 3.0 ±0.2 | 0.99 |
| dry season | 1430±24 | 0.07±0.01 | 1.8 ±0.1 | 0.99 |

Third, we added the results from the seasonally resolved Twomey and erf fits to the overview Table 3.

Fourth, we modified the corresponding text section in Sect. 3.5.3 from:

"Figure 11a and b show the corresponding $N_{CCN,p}(S)$ *versus* $N_{CCN}(S)$ scatter plots. In general, parametrizations based on CCN spectra yield a mean state based on average concentrations (see fit parameters in Fig. 10) and ignore the temporal variability of the aerosol concentrations (Martins et al., 2009; Rose et al., 2010; Jurányi et al., 2011). On closer inspection, Table 3 shows that the erf fit allows somewhat better predictions (deviation of power law fit about 227 % *versus* 215 % for erf fit), which can be explained by the fact that the erf fit presents the experimental data more appropriately (compare Fig. 10). Overall, however, the power law fit and the erf fit approaches give rather poor correlations, due to the missing representation of the aerosol's temporal variability, which is an inherent limitation of the CCN spectra parametrization. It can be concluded that this parametrization requires a minimum of aerosol input data (i.e., only the parameters of the corresponding fit function), which explains its wide distribution in various modelling attempts. However, Fig. 10 and Table 3 show that this simplicity is clearly at the expense of the prediction accuracy."

to:

"Figure 11a and b show the corresponding $N_{CCN,p}(S)$ *versus* $N_{CCN}(S)$ scatter plots based on the annual mean CCN spectrum, using the Twomey and erf fits. In general, parametrizations based on CCN spectra yield a mean state based on average concentrations (see fit parameters in Fig. 10 as well as Table 5 and 6) and ignore the temporal variability of the aerosol's abundance (Martins et al., 2009; Rose et al., 2010; Jurányi et al., 2011). Table 3 shows that the erf fit allows somewhat better predictions (e.g., deviation of power law fit about 227 % *versus* 215 % for erf fit in case of annual mean and 80 % *versus* 75 % for the seasonally

resolved case), which can be explained by the fact that the erf fit presents the experimental data more appropriately (compare with Fig. 10). Overall, however, the power law and erf fit approaches give rather poor correlations due to the missing representation of the aerosol's temporal variability. This is particularly obvious for the annual mean case since the total aerosol abundance varies significantly between wet and dry season conditions. Accordingly, the CCN spectra parametrization, which operates with constants, predictably underestimates the dry season conditions and overestimates the wet season conditions. In addition to the analytical fit approaches for the *annual mean* spectrum (Fig. 11a and b) we conducted an analogous CCN prediction based on *seasonally* resolved CCN spectra (Fig. 11c and d). The prediction accuracy clearly improves (e.g., deviation of erf fit for annual mean case equals 215 % *versus* 75 % for seasonally resolved case; see Table 3). Figure 11 illustrates that the prediction accuracy of parametrizations, which rely on analytical fit functions of CCN spectra (i.e., Twomey, erf, and related functions), improves with decreasing variability of the aerosol population (e.g., for shorter periods with less variable aerosol properties). However, the missing representation of the aerosol's temporal variability remains to be an *inherent limitation* of the CCN spectra parametrization. It can be concluded that this parametrization requires a minimum of aerosol input data (i.e., only the parameters of the corresponding fit function), which explains its wide distribution in various modelling attempts. However, Fig. 11 and Table 3 show that this simplicity is clearly at the expense of the prediction accuracy."

[2.5]     Referee comment: Technical corrections: The text begins expressing supersaturation by "supersaturation S". Then it changes to "S", then to "S levels". Is it correct? Please check it.

Author Response: This is correct. On page 7 in line 11, we introduced the symbol $S$ for supersaturation. Throughout the text we then only refer to "$S$" or to "$S$ levels", which is synonymously used for "supersaturations".

Page 1: The title has been changed from
   Long-term observations of atmospheric aerosol, cloud condensation nuclei concentration and hygroscopicity in the Amazon rain forest – Part 1: Size-resolved characterization and new model parameterizations for CCN prediction

to:

[revised manuscript text omitted]

Page 14, line 32: The following sentence has been inserted:
For comparison, the diurnal cycles in $N_\mathrm{CN}$ concentration have been added to Fig. 7, which confirm the absence of strong diurnal variations in the aerosol population.

Page 17, line 24: The following section has been changed from:
The obtained fit parameters $N\mathrm{CCN}(1\%) = 998$ cm-3 (sometimes also called $c$) and $k = 0.36$ agree with results from previous measurements that are summarized by Martins et al. (2009b). The power law function has become a widely used parametrization due to its simplicity (Cohard et al., 1998). However, it is based on strong assumptions as well as not related to the physical basis of the fitted data and thus reveals certain drawbacks, such as the poor representation of $N\mathrm{CCN}(S)$ at small $S$ (i.e., $< 0.2$ %) as well as the fact that for larger $S$ (i.e., $> 1.2$ %) it does not converge against $N\mathrm{CN}$ which is, for physical reasons, the upper limit.

to:

Besides the annual mean spectrum, we also conducted a Twomey fit for the seasonally resolved CCN spectra (not shown) and summarized the resulting fit parameters in Table 5. The obtained fit parameters (e.g., for the annual mean CCN spectrum) $N_\mathrm{CCN}(1\%) = 998$ cm$^{-3}$ (sometimes also called $c$) and $k = 0.36$ agree with results from previous measurements that are summarized by

Martins et al. (2009b). The power law function has become a widely used parametrization due to its simplicity (Cohard et al., 1998). However, because it is based on strong assumptions and not related to the physical basis of the fitted data, it has certain drawbacks, such as the poor representation of $N_{CCN}(S)$ at small $S$ (i.e., < 0.2 %), as well as the fact that for larger $S$ (i.e., > 1.2 %) it does not converge against $N_{CN}$, which is, for physical reasons, the upper limit.

Page 18, line 6:The following sentence has been inserted:

For this approach, we also summarized the corresponding fit parameters for the annual mean CCN spectrum and the seasonally resolved cases in Table 6.

Page 18, line 7: The following section has been changed from:

[revised manuscript text omitted]